# SpinQuant: LLM Quantization with Learned Rotations

**Zechun Liu**[*]   **Changsheng Zhao**[*]   **Igor Fedorov**   **Bilge Soran**   **Dhruv Choudhary**
**Raghuraman Krishnamoorthi**   **Vikas Chandra**   **Yuandong Tian**   **Tijmen Blankevoort**
Meta

## Abstract

Post-training quantization (PTQ) techniques applied to weights, activations, and the KV cache greatly reduce memory usage, latency, and power consumption of Large Language Models (LLMs), but may lead to large quantization errors when outliers are present. Rotating activation or weight matrices helps remove outliers and benefits quantization. In this work, we identify a collection of applicable rotation parameterizations that lead to identical outputs in full-precision Transformer architectures while enhancing quantization accuracy. In addition, we find that some random rotations lead to much better quantization than others, with an up to *13 points* difference in downstream zero-shot reasoning performance. As a result, we propose `SpinQuant`, a novel approach that incorporates learned rotation matrices for optimal quantized network accuracy. With 4-bit quantization of weight, activation, and KV-cache, `SpinQuant` narrows the accuracy gap on zero-shot reasoning tasks with full precision to merely 2.9 points on the LLaMA-2 7B model, surpassing LLM-QAT by 19.1 points and SmoothQuant by 25.0 points. Furthermore, SpinQuant also outperforms concurrent work QuaRot, which applies random rotations to remove outliers. In particular, for LLaMA-3 8B models that are hard to quantize, `SpinQuant` reduces the gap to full precision by up to 45.1% relative to QuaRot. Code is available at `github.com/facebookresearch/SpinQuant`.

## 1 Introduction

Large Language models (LLMs) have demonstrated impressive performance across many disciplines. SoTA open source models (*e.g.*, LLaMA (Touvron et al., 2023b), Mistral (Jiang et al., 2023), etc) and proprietary LLMs (*e.g.*, GPT (Achiam et al., 2023), Gemini(Team et al., 2023), etc) have been used in general purpose chatting assistants, medical diagnosticians (Thirunavukarasu et al., 2023), computer game content generators (Cox and Ooi, 2023), coding co-pilots (Roziere et al., 2023), and much more.

To serve such a high demand, the inference cost becomes a real issue. Many effective techniques have been developed. Post-training Quantization (PTQ), as one effective category of techniques, quantizes the weights (or activations) into low-precision and thus reduces the memory usage and may significantly improve latency. This is not only important for server-side inference, but also for on-device scenarios with small-sized LLMs (Liu et al., 2024; AI@Meta, 2024).

When applying quantization, outliers remain an open challenge because they stretch the quantization range, leaving fewer effective bits available for the majority of values. Prior research mitigates this challenge by trading quantization difficulty between weights and activations (Xiao et al., 2022; Lin et al., 2023) or employing mixed-precision to handle outliers (Zhao et al., 2023). In this work, we focus on a new angle: multiplying the weight matrix with a rotation matrix to reduce outliers and enhance quantizability. Inspired by (Elhage et al., 2023) and SliceGPT (Ashkboos et al., 2023a), we leverage the property of rotational invariance to construct rotation matrices in pairs from identity mapping, which can be integrated into nearby weights without affecting the overall network outputs. By applying these random rotations, we produce a distribution of weight or activation entries that is outlier-less, facilitating easy quantization.

---

[*] Equal contribution. Correspondence to: Zechun Liu <zechunliu@meta.com>.

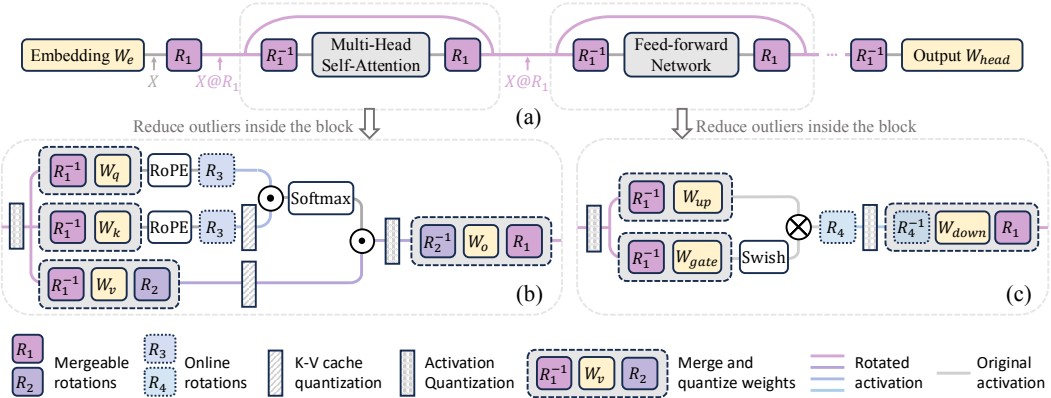

Figure 1: **Overall diagram of rotation.** (a) The residual stream can be rotated in the transformer network, resulting in numerically equivalent floating point networks before and after rotation. The rotated activations exhibit fewer outliers and are easier to quantize. (b) & (c) The rotation matrix can be integrated with the corresponding weight matrices and we further define $R_2$, $R_3$, and $R_4$ for reducing outliers inside the block.

In addition to using random rotation, which statistically works well, we find that the performance of quantized network could *vary a lot* with different rotation matrices. For example, the downstream averaged accuracy on zero-shot reasoning tasks may change up to 13 points with different rotations. As a result, we propose SpinQuant that *integrates* and *optimizes* the rotation matrix to minimize the final loss of the quantized network, with fixed weight parameters, by employing the *Cayley SGD* (Li et al., 2020), a proficient technique for optimizing orthonormal matrices. This optimization does not alter the full-precision network output but refines the intermediate activations and weights, making them more quantization-friendly.

In SpinQuant, we introduce two rotation strategies tailored for different complexity levels: $\text{SpinQuant}_{no\ had}$ and $\text{SpinQuant}_{had}$. Here, *had* refers to hadamard rotation matrix. In $\text{SpinQuant}_{no\ had}$, as depicted in Figure 1(b), we implement shortcut rotation ($R_1$) and $W_v$-$W_o$ pair rotation ($R_2$), which can be directly absorbed into the respective weight matrices. During inference, the original weights are simply replaced with the rotated quantized weights, eliminating the need for modification in the forward pass. Conversely, in $\text{SpinQuant}_{had}$, designed for scenarios with low-bit quantization of KV cache or activations (e.g., 4-bit), we further incorporate online Hadamard rotation matrices ($R_3$, $R_4$) to address activation outliers inside MLP block and KV cache.

To rigorously assess the effectiveness of SpinQuant, we executed comprehensive experiments across seven leading Large Language Models (LLMs), including LLaMA-2(Touvron et al., 2023b) models (7B/13B/70B), LLaMA-3(AI@Meta, 2024) models (1B/3B/8B), and the Mistral (Jiang et al., 2023) 7B model. The key contributions of this study are summarized as follows:

- We introduce SpinQuant, the first method that employs learned rotations to mitigate outliers in weight and activation distributions, boosting the performance of quantized LLMs.

- We reveal that random rotations introduce substantial variance in quantized network performance. We propose optimizing rotation matrices within *Stiefel* manifold, directly minimizing the final loss of rotated quantized network. Ablation studies validate that our learned rotations consistently outperform random rotations, with improvements up to 16.2 points.

- $\text{SpinQuant}_{no\ had}$ merges rotation matrices into pre-trained weights without altering the network architecture, significantly narrowing the W4A8KV8 quantization performance gap from 12.1 to 1.6 on the Mistral-7B model in zero-shot commonsense reasoning tasks. Noteworthily, $\text{SpinQuant}_{no\ had}$ W4A8 quantization achieves comparable performance as state-of-the-art weight only quantization methods like QuIP# (Tseng et al., 2024) and OminiQuant (Shao et al., 2023) on LLaMA-2.

- $\text{SpinQuant}_{had}$ attains an average accuracy of 64.0 in extreme W4A4KV4 quantization settings on LLaMA-2 7B. This represents a mere 2.9 point gap from the full-precision network, a substantial improvement over the previous LLM-QAT (Liu et al., 2023c) approach, which exhibited a 22.0 point gap under identical precision conditions.

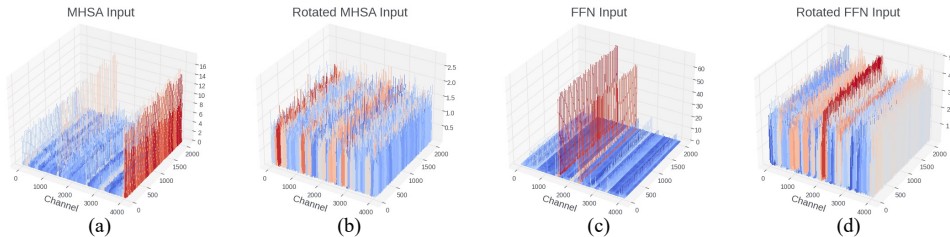

Figure 2: Activation distribution in LLaMA-2 7B model before and after rotation. Outliers exist in particular channels before rotation. Since channel-wise quantization is not supported in most hardware, outlier removal using rotation enables accurate token-wise or tensor-wise quantization.

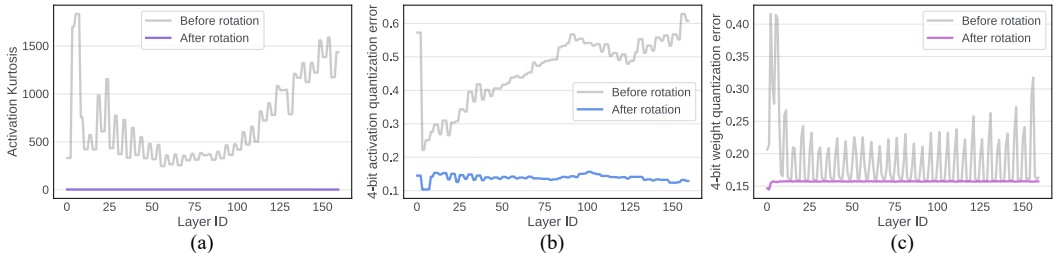

Figure 3: Outlier measurement and quantization error across input activation and weights in the five layers that take inputs from the residual (Q/K/V/Up/Gate-projection) of each block in the LLaMA-2 7B model. (a) After rotation, *kurtosis* of activation distributions is significantly reduced to approximately three across all layers. Quantization error is reduced after rotation in both (b) activations and (c) weights.

## 2 MOTIVATION

Quantization reduces the precision of weights (and/or activations) in a neural network in order to save memory and lower the latency. The quantization process can be formulated as:

$$X_Q = \alpha \lfloor \frac{X_R - \beta}{\alpha} \rceil + \beta \tag{1}$$

where $\alpha = \frac{\max(|X_R|)}{2^{N-1}-1}, \beta = 0$ in symmetric quantization or $\alpha = \frac{\max(X_R)-\min(X_R)}{2^N-1}, \beta = \min(X_R)$ in asymmetric quantization. Here $X_Q$ is a quantized tensor and $X_R$ is a real-valued FP16 tensor. $N$ is number of bits. For Large language models (LLMs), the presence of outliers extends the range of weight/activation values and increases the reconstruction errors for normal values (Dettmers et al., 2022; Liu et al., 2023b; Yelysei Bondarenko, 2023) (Figures 2 (a)&(c)).

### 2.1 OUTLIER REDUCTION

There exist many ways to mitigate the effect of outliers (Xiao et al., 2022; Dettmers et al., 2022). In this paper, we propose to use optimized rotation to reduce outliers. Intuitively, a random rotation matrix statistically blends large and small weights together into a well-behaved distribution with fewer outliers (Elhage et al., 2023), and thus is easier to quantize.

Figure 3 (a) illustrates the measurement of the *Kurtosis* $\kappa$ of the activations before and after rotation. $\kappa$ quantifies the "*tailedness*" of a real-valued random variable's probability distribution. A larger $\kappa$ indicates more outliers, while $\kappa \approx 3$ suggests a Gaussian-like distribution. In Figure 3 (a), the activation distribution in the transformer contains numerous outliers, with $\kappa$ of many layers exceeding 200. However, after multiplying these activations with a random rotation matrix, the $\kappa$ across all layers becomes approximately 3, indicating a more Gaussian-shaped distribution that is easier to quantize. This is corroborated by Figure 3 (b), where the quantization error of the activation tensor significantly decreases after rotation.

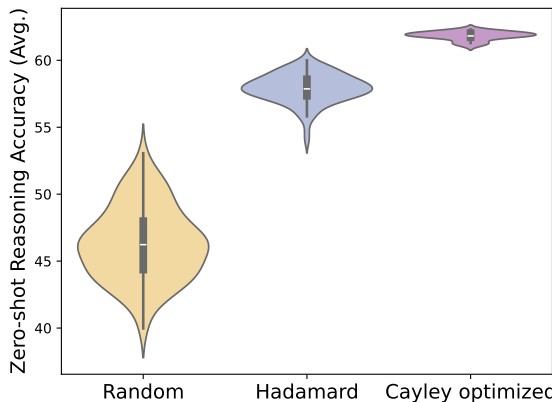

Figure 4: The performance distributions of W4A4 quantized LLaMA-2 7B under different random rotations, using network-level parameterization (Sec. 3.1). We compare the distributions using random floating-point rotations, random Hadamard matrices, and optimized rotation matrices with *Cayley* optimization (Sec. 3.2). Despite that Hadamard matrices mostly perform better than random rotations, both random groups demonstrate large variance. In contrast, by optimizing the rotation matrix with *Cayley* optimization (*i.e.*, `SpinQuant`), the performance is improved significantly and the variance becomes much smaller.

## 2.2 RANDOM ROTATIONS PRODUCE LARGE VARIANCE

Interestingly, while statistically random rotation leads to better quantization, not all random rotations give the same quantization outcome. To show this, we tested the zero-shot average accuracy of the rotated version of LLaMA-2 7B, quantized to 4-bit weight and 4-bit activation, under 100 randomized trials. As shown in Figure 4, the performance variance is substantial, with the best random rotation matrix outperforming the worst by 13 points. Random Hadamard matrices [1] outperform random rotation matrices, in consistent with the findings in (Tseng et al., 2024) that Hadamard matrices yield tighter bounds on weight maximal value. However, even random Hadamard rotation matrices exhibit a non-negligible variance in final performance, as large as 6 points.

Given the huge variance across multiple trials of rotations, a natural question arises: *Is it possible to optimize the rotation to maximize the benefit of quantization?* We affirmatively answer this question by presenting a viable framework with quantization-oriented rotation learning that consistently achieves high accuracy across 7 models and 4 low-bit quantization settings.

## 3 METHOD

In this section, we introduce `SpinQuant`, a framework that integrates and optimizes rotations in LLMs targeting at quantization loss. We start with defining rotation parameterization of popular LLM architectures, which includes two mergeable rotation matrices ($R_1$, $R_2$) that produce rotationally invariant full-precision network, and two online Hadamard rotation ($R_3$, $R_4$) to further reduce the outliers for extreme activation and KV-cache quantization. Then, we present how to optimize these rotation matrices on *Stiefel* manifold with target loss.

### 3.1 ROTATION PARAMETERIZATION

**Rotating activations in residual** As shown in Figure 1(a), we rotate the activations in the residual path by multiplying the embedding output $X$ with a random rotation matrix ($R_1$). This rotation removes outliers and eases the quantization of the input activations to the fully-connected layers that read from the residual. To maintain numerical invariance, we reverse the rotation of the activation by

---

[1] A Hadamard matrix $H$ is a special type of rotation matrix, where the entries of the matrix are solely $\pm\sqrt{n}$. Given a Hadamard matrix $H$, we can generate $2^n$ different random Hadamard matrices by multiplying with $S$, a diagonal matrix with elements $s_i$ randomly chosen from $\{-1, 1\}$.

multiplying it with $R_1^T (= R_1^{-1})$ prior to its passage through the attention block and feed-forward network, which contains non-linearity. When the quantization is not present, the full-precision network remains intact no matter which rotation is applied.[2] The rotation matrices can be merged into corresponding weight matrices, as illustrated in Figures 1(b)&(c). After absorption, no new parameters are introduced in the network. We can now modify $R_1$ freely without impacting the floating-point network's accuracy or parameter count.

**Rotating activations in the attention block** As depicted in Figure 1(b), in the attention block, we propose to rotate the value matrix by multiplying $R_2$, and the activations to out-projection layer by $R_2^T$ head-wisely. $R_2$ has the shape of $(D_{head}, D_{head})$ and can be independently chosen across layers. The numerical in-variance is illustrated in Figure 5, these two rotations can be offset in a full-precision network since there are no operators between $R_2$ and $R_2^T$. Meanwhile, it can improve quantization for value cache and input activations to out-projection layer without introducing any new parameters in the network.

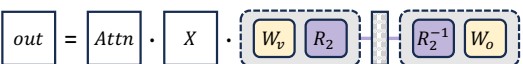

Figure 5: Rotation equivalence in Multi-Head Self-Attention.

We denote the method with only $R_1$ and $R_2$ inserted and optimized as SpinQuant$_{no\ had}$, which can readily achieve significant accuracy improvement than previous quantization methods, and closing the gap between W4A8 quantized LLMs and their full-precision counterparts to $0.1 - 2.5$ points on zero-shot commonsense reasoning averaged accuracy.

**Additional unabsorbed rotations** To further enhance outlier suppression for lower-bit (*e.g.* 4-bit) activation quantization, we incorporate a Hadamard matrix multiplication ($R_4$ in Figure 1(c)) inside the feed-forward block, reducing the outliers in the input to the down projection layer, similar to (Tseng et al., 2024; Ashkboos et al., 2023b). Hadamard rotation can be computed with fast hadamard transform and introduce marginal overhead to the inference latency. Similarly, Hadamard matrix ($R_3$ in Figure 1(b)) can be inserted when low-bit KV cache quantization is required. We denote the resulting method, equipped with all rotations, as SpinQuant$_{had}$. Next, we demonstrate how to jointly optimize these rotations.

### 3.2 *Cayley*-OPTIMIZED ROTATION

As illustrated in Figure 1, we have determined that the incorporation of four rotation matrices ($R_1$, $R_2$, $R_3$, $R_4$) can improve quantization performance while preserving numerical consistency in a full-precision network. Given that $R_3$ and $R_4$ are online rotation operations, meaning they cannot be absorbed into the weight matrix, we retain them as Hadamard matrices. This is because online Hadamard transforms can be efficiently implemented without significant overhead. We then define the optimization objective as identifying the optimal rotation matrix $R_1$ and $R_2$ that minimizes the final loss of the quantized network:

$$\underset{R \in \mathcal{M}}{\arg\min} \mathcal{L}_Q(R_1, R_2 \mid W, X) \tag{2}$$

Here, $\mathcal{M}$ represents the *Stiefel* manifold *i.e.*, the set of all orthonormal matrices. $\mathcal{L}_Q(\cdot)$ denotes the task loss, such as cross-entropy, on the calibration set. It is a function of $\{R_1, R_2\}$, given the fixed pretrained weights $W$ and the input tensor $X$ and with the quantization function $Q$ in the network. To optimize the rotation matrix on the *Stiefel* manifold, we employ the *Cayley SGD* method (Li et al., 2020), which is an efficient optimization algorithm on the *Stiefel* manifold. More specifically, in each iteration, the update of the rotation $R$ is parameterized as the following:

$$R' = \Delta R(Y)R := \left(I - \frac{\alpha}{2}Y\right)^{-1}\left(I + \frac{\alpha}{2}Y\right)R \tag{3}$$

---

[2]In a pre-norm LLM like LLaMA (Touvron et al., 2023a), we can convert a transformer network into a rotation-invariant network by incorporating the RMSNorm scale parameters $\alpha$ into the weight matrix right after the RMSNorm layer (Ashkboos et al., 2023a).

where $\Delta R(Y) := (I - \frac{\alpha}{2}Y)^{-1}(I + \frac{\alpha}{2}Y)$ is the *Cayley Transform* of a skew-symmetric matrix $Y$ (*i.e.*, $Y^\top = -Y$). $Y$ is computed from a projection $\hat{G}$ of the gradient $G := \nabla_R \mathcal{L}_Q$ of the loss function:

$$Y = \hat{G} - \hat{G}^\top, \qquad \hat{G} := GR^\top - \frac{1}{2}RR^\top GR^\top \tag{4}$$

It can be shown that $\Delta R(Y)$ is always orthonormal and thus $R'$ is guaranteed to be orthonormal ($R'^\top R' = I$) if $R$ is orthonormal. While Eqn. 3 requires a matrix inverse, the new rotation matrix $R'$ can be computed via an efficient fixed point iteration (Li et al., 2020). Overall, the approach maintains the property of orthonormality with only $\sim$2 times the computation time per iteration compared to a naive SGD algorithm.

We apply the *Cayley SGD* method to solve Eqn. 2 for $\{R_1, R_2\}$, while the underlying weight parameters in the network remain frozen. $\{R_1, R_2\}$ count for only $\sim$0.26% of the weight size and is constrained to be orthonormal. Consequently, the underlying floating-point network remains unchanged, and the rotation only influences the quantization performance.

By employing *Cayley* optimization to update the rotation for 100 iterations on an 800-sample WikiText2 calibration dataset, we obtain a rotation matrix that outperforms the best random matrix and random Hadamard matrix in 100 random seeds, shown in Figure 4. The *Cayley*-optimized rotation exhibits minimal variance when initiated from different random seeds. The rotation matrices are initialized with random Hadamard matrices for optimization and our ablation study in Section 4.3.3 demonstrates that the optimized rotation is robust to random rotation initialization as well.

# 4 EXPERIMENTS

We conduct experiments on the LLaMA-2 (Touvron et al., 2023b) models (7B/13B/70B), LLaMA-3 (AI@Meta, 2024) models (1B/3B/8B) and Mistral (Jiang et al., 2023) 7B model. Our evaluation of the proposed `SpinQuant` was carried out on eight zero-shot commonsense reasoning tasks. These tasks include BoolQ (Clark et al., 2019), PIQA (Bisk et al., 2020), SIQA (Sap et al., 2019), HellaSwag (Zellers et al., 2019), WinoGrande (Sakaguchi et al., 2021), ARC-easy and ARC-challenge (Clark et al., 2018), and OBQA (Mihaylov et al., 2018). Additionally, we also report the perplexity score on WikiText2 testset (Merity et al., 2016) for our evaluation.

## 4.1 EXPERIMENTAL SETTINGS

We employ *Cayley SGD* (Li et al., 2020) to optimize the rotation matrix, $R_1$ and $R_2$, both initialized as a random Hadamard matrix, while maintaining all network weights constant. $R_1$ is the residual rotation, shaped as $(D_{token}, D_{token})$. $R_2$ is head-wise rotation in each attention block, shaped as $(D_{head}, D_{head})$ and is separately learned in each layer. The learning rate starts at 1.5 and linearly decays to 0. We utilize 800 samples from WikiText-2 to optimize rotation for 100 iterations. It takes only $\sim$ 13 / 18 / 30 minutes for LLaMA-3 1B / 3B / 8B, respectively, and $\sim$ 25 / 30 minutes for LLaMA-2 7B / 13B, respectively. For LLaMA-2 70B, it takes $\sim$ 3.5 hours and for Mistral-7B it takes $\sim$ 16 minutes.

In the main results, we optimize the rotation with respect to the activation quantized network, where the weights remain 16-bit. After rotation is learned, we apply GPTQ on the rotated weights (Frantar et al., 2022), for which we adhere to the standard GPTQ settings by using 128 samples from WikiText-2 with a sequence length of 2048 as the calibration set for GPTQ quantization. In the main table, we present the results of `SpinQuant` with GPTQ, and in the ablation study, while we also show the results of employing simple round-to-nearest (RTN) quantization in the ablation study.

## 4.2 MAIN RESULTS

We present two rotation schemes `SpinQuant`$_{no\ had}$ and `SpinQuant`$_{had}$ to accommodate different scenarios. In Table 1, we use seven models and four most commonly used bit-width settings to provide a guideline on which rotation scheme should be chosen in practice.

Recap `SpinQuant`$_{no\ had}$ uses learned rotation $R_1$ and $R_2$ only, which can be merged into corresponding model weights during inference time after the rotation is learned. Using `SpinQuant`$_{no\ had}$

Table 1: Comparison of the perplexity score on WikiText2 and averaged accuracy on eight Zero-shot Common Sense Reasoning tasks. Results for SmoothQuant (Xiao et al., 2022), LLM-QAT (Liu et al., 2023c), GPTQ (Frantar et al., 2022) were obtained using their publicly released codebase. While OmniQuant (Shao et al., 2023), AWQ (Lin et al., 2023), and QuIP# (Tseng et al., 2024) results were quoted from their papers. Full results are in the Appendix.

| #Bits (W-A-KV) | Method | LLaMA-2 7B 0-shot[8] Avg.(↑) | Wiki (↓) | LLaMA-2 13B 0-shot[8] Avg.(↑) | Wiki (↓) | LLaMA-2 70B 0-shot[8] Avg.(↑) | Wiki (↓) | LLaMA-3.2 1B 0-shot[8] Avg.(↑) | Wiki (↓) | LLaMA-3.2 3B 0-shot[8] Avg.(↑) | Wiki (↓) | LLaMA-3 8B 0-shot[8] Avg.(↑) | Wiki (↓) | Mistral-7B 0-shot[8] Avg.(↑) | Wiki (↓) |
|---|---|---|---|---|---|---|---|---|---|---|---|---|---|---|---|
| 16-16-16 | FloatingPoint | 66.9 | 5.5 | 68.3 | 5.0 | 72.9 | 3.3 | 56.9 | 13.4 | 63.9 | 10.7 | 69.6 | 6.1 | 71.0 | 5.4 |
| 4-8-16 | RTN | 62.4 | 7.9 | 57.3 | 6.7 | 68.6 | 5.0 | 55.4 | 20.7 | 58.6 | 29.0 | 65.5 | 8.2 | 59.3 | 6.8 |
|  | SmoothQuant | 58.9 | 7.5 | 63.6 | 6.1 | 70.6 | 4.1 | 47.1 | 1e2 | 55.6 | 3e2 | 61.0 | 10.7 | – | – |
|  | LLM-QAT | 64.8 | 11.4 | 67.5 | 14.5 | – | – | 53.2 | 21.0 | 60.8 | 41.1 | 67.2 | 7.7 | – | – |
|  | AWQ (w4) | – | 6.2 | – | 5.1 | – | – | – | – | – | – | – | – | – | – |
|  | OmniQuant (w4) | – | 5.7 | – | 5.0 | – | 3.5 | – | – | – | – | – | – | – | – |
|  | QuIP# (w4) | – | 5.6 | – | 5.0 | – | 3.4 | – | – | – | – | – | – | – | – |
|  | GPTQ | 64.9 | 20.2 | 65.2 | 5.9 | 71.7 | 4.3 | 55.0 | 17.3 | 58.7 | 25.2 | 64.5 | 7.2 | 51.7 | 8.6 |
|  | SpinQuant$_{no\ had}$ | 65.7 | 5.8 | 68.2 | 5.1 | 72.1 | 3.7 | 56.0 | 15.3 | 61.4 | 11.6 | 68.6 | 6.7 | 68.8 | 5.7 |
|  | SpinQuant$_{had}$ | 65.7 | 5.7 | 68.1 | 5.0 | 72.7 | 3.5 | 56.5 | 14.4 | 63.2 | 11.5 | 68.4 | 6.5 | 69.9 | 5.5 |
| 4-8-8 | RTN | 62.5 | 7.9 | 57.6 | 6.7 | 68.4 | 5.0 | 55.7 | 20.7 | 58.4 | 28.8 | 65.3 | 8.2 | 58.9 | 6.7 |
|  | SmoothQuant | 58.8 | 7.5 | 63.4 | 6.1 | 70.5 | 4.1 | 47.1 | 1e2 | 55.5 | 3e2 | 60.9 | 10.7 | – | – |
|  | LLM-QAT | 64.6 | 11.4 | 67.5 | 14.2 | – | – | 53.1 | 21.0 | 60.5 | 39.3 | 66.9 | 7.6 | – | – |
|  | GPTQ | 64.8 | 20.2 | 65.3 | 5.9 | 71.6 | 4.3 | 54.8 | 17.3 | 58.7 | 24.1 | 64.6 | 7.2 | 51.7 | 8.6 |
|  | SpinQuant$_{no\ had}$ | 65.8 | 5.8 | 68.1 | 5.1 | 72.2 | 3.7 | 55.7 | 15.3 | 61.8 | 11.7 | 68.6 | 6.7 | 69.4 | 5.7 |
|  | SpinQuant$_{had}$ | 65.8 | 5.7 | 68.2 | 5.1 | 72.7 | 3.5 | 55.8 | 14.3 | 63.2 | 11.2 | 68.8 | 6.5 | 70.2 | 5.5 |
| 4-4-16 | RTN | 35.6 | 2e3 | 35.3 | 7e3 | 35.1 | 2e5 | 41.2 | 1e2 | 42.1 | 7e2 | 43.9 | 2e2 | 41.4 | 4e2 |
|  | SmoothQuant | 41.8 | 3e2 | 44.9 | 34.5 | 57.7 | 57.1 | 37.9 | 2e3 | 43.6 | 4e2 | 40.3 | 9e2 | – | – |
|  | LLM-QAT | 47.8 | 12.9 | 34.3 | 4e3 | – | – | 42.0 | 62.1 | 46.9 | 37.6 | 44.9 | 42.9 | – | – |
|  | GPTQ | 36.8 | 9e3 | 35.2 | 5e3 | 35.5 | 2e6 | 41.6 | 1e2 | 43.4 | 3e2 | 40.6 | 2e2 | 40.4 | 3e2 |
|  | SpinQuant$_{no\ had}$ | 57.0 | 9.2 | 61.8 | 7.2 | 61.0 | 7.3 | 44.8 | 48.4 | 52.9 | 22.4 | 51.9 | 18.6 | 52.7 | 13.4 |
|  | SpinQuant$_{had}$ | 64.1 | 5.9 | 67.2 | 5.2 | 71.0 | 3.8 | 53.5 | 15.3 | 61.0 | 11.1 | 65.8 | 7.1 | 68.4 | 5.7 |
| 4-4-4 | RTN | 37.1 | 2e3 | 35.5 | 7e3 | 35.0 | 2e5 | 40.6 | 2e2 | 41.2 | 8e2 | 43.1 | 3e2 | 41.4 | 4e2 |
|  | SmoothQuant | 39.0 | 7e2 | 40.5 | 56.6 | 55.9 | 10.5 | 36.5 | 2e3 | 40.0 | 6e2 | 38.7 | 2e3 | – | – |
|  | LLM-QAT | 44.9 | 14.9 | 35.0 | 4e3 | – | – | 41.5 | 76.2 | 45.9 | 42.0 | 43.2 | 52.5 | – | – |
|  | GPTQ | 36.8 | 9e3 | 35.2 | 5e3 | 35.6 | 1e6 | 41.6 | 1e2 | 41.1 | 4e2 | 40.5 | 2e2 | 41.3 | 2e2 |
|  | SpinQuant$_{no\ had}$ | 56.0 | 9.2 | 60.7 | 7.1 | 62.0 | 7.4 | 45.3 | 47.7 | 52.9 | 22.4 | 52.6 | 18.6 | 52.4 | 13.7 |
|  | SpinQuant$_{had}$ | 64.0 | 5.9 | 66.9 | 5.3 | 71.2 | 3.8 | 53.4 | 15.9 | 60.5 | 11.4 | 65.5 | 7.3 | 68.6 | 5.8 |

only needs to replace the original model weights with the rotated model weights, necessitating no modification to the forward pass nor any additional kernel support. While SpinQuant$_{had}$ comprises both learned rotations ($R_1$, $R_2$) and the online Hadamard rotations ($R_3$, $R_4$). During inference time, $R_3$ and $R_4$ can be computed with fast Hadamard kernel (Tseng et al., 2024) and we show in Sec. 4.5, the online Hadamard rotation only introduces ∼8% of the network latency overhead.

As shown in Table 1, in the scenarios where weights are quantized to 4-bit and activations are quantized to 8-bit, using SpinQuant$_{no\ had}$ can readily achieve good performance. For example, SpinQuant$_{no\ had}$ enhances the 4-8-8 quantized Mistral 7B by 10.5 points. In llama3-8B, SpinQuant$_{no\ had}$ achieves more than 4.1 point improvements compared to GPTQ (Frantar et al., 2022) on 4-8-16 setting, and leaving the gap to full-precision network to only 1.0 point. In these settings with activations not extremely quantized, using SpinQuant$_{no\ had}$ is a viable solution, and adding additional online Hadamard rotation yields marginal benefit.

In contrast, when activations are quantized to 4 bits, the accuracy drops significantly and most previous methods fail to produce meaningful results. SpinQuant$_{no\ had}$ bridge the gap by up to 20 points. In 4-4-4 quantized LLaMA-2 models, SpinQuant$_{no\ had}$ significantly surpasses LLM-QAT (Liu et al., 2023c), by 11.1 points on 7B model and outperforms SmoothQuant (Xiao et al., 2023) by 20.2 on the 13B model, thereby reducing the gap to the corresponding full-precision network from 22.0 / 27.8 points to 10.9 / 7.6 points respectively. Still, the gap to the full-precision network is non-negligible. In this scenario, SpinQuant$_{had}$ can further improve the accuracy by more than 5 points and close the gap to the respective FP network to 2-4 points. In 4-4-4 quantized LLaMA-2 7B/13B/70B models, SpinQuant$_{had}$ leaves only a 2.9/1.4/1.7 accuracy gap to the corresponding full-precision network, significantly surpassing the previous SoTA methods by 19.1/16.4/15.3 points, respectively.

In addition, compared to the state-of-the-art weight-only quantization methods, OmniQuant (Shao et al., 2023), AWQ (Lin et al., 2023) and QuIP# (Tseng et al., 2024), SpinQuant achieves similar evaluation perplexity on Wiki dataset with 4-bit weights and 8-bit activations, and without using advance vector quantization technique. These results show SpinQuant is suitable for various scenarios and achieves state-of-the-art performance.

Table 2: Compared to Hadamard rotation, `SpinQuant` learned rotation consistently outperform by a significant margin. Results are averaged accuracy on eight Zero-shot CommonSense Reasoning tasks.

| | LLaMA-3.2 3B | | LLaMA-3 8B | | Mistral-7B | |
|---|---|---|---|---|---|---|
| | 4-4-16 | 4-4-4 | 4-4-16 | 4-4-4 | 4-4-16 | 4-4-4 |
| Random Hadamard $R_{\{1,2\}}$ | 49.8 | 49.6 | 49.5 | 50.0 | 51.4 | 51.5 |
| `SpinQuant`$_{no\ had}$ $R_{\{1,2\}}$ | $52.9_{(\uparrow 3.1)}$ | $52.9_{(\uparrow 3.3)}$ | $51.9_{(\uparrow 2.4)}$ | $52.6_{(\uparrow 2.5)}$ | $52.7_{(\uparrow 1.3)}$ | $52.4_{(\uparrow 0.9)}$ |
| Random Hadamard $R_{\{1,2,3,4\}}$ | 59.0 | 58.4 | 64.2 | 63.9 | 52.7 | 52.4 |
| `SpinQuant`$_{had}$ $R_{\{1,2,3,4\}}$ | $61.0_{(\uparrow 2.1)}$ | $60.5_{(\uparrow 2.2)}$ | $65.8_{(\uparrow 1.6)}$ | $65.5_{(\uparrow 1.6)}$ | $68.4_{(\uparrow 15.7)}$ | $68.6_{(\uparrow 16.2)}$ |

Table 3: Ablation study on compatibility with GPTQ (Frantar et al., 2022) on a LLaMA2-7B model.

| #Bits(W-A-KV) | Task | *Cayley* on 4-4-KV | *Cayley* on 16-4-KV |
|---|---|---|---|
| 4-4-16 | 0-shot[8] Avg. | $61.0_{\pm 1.0}$ | $64.1_{\pm 0.4}$ |
| | Wiki | $6.7_{\pm 0.07}$ | $5.9_{\pm 0.00}$ |
| 4-4-4 | 0-shot[8] Avg. | $60.9_{\pm 0.6}$ | $64.0_{\pm 0.3}$ |
| | Wiki | $6.8_{\pm 0.15}$ | $5.9_{\pm 0.01}$ |

## 4.3 ABLATION STUDIES

### 4.3.1 LEARNED ROTATION VS RANDOM ROTATION

In Table 2, we contrast the use of random Hadamard rotations with `SpinQuant`'s optimized rotations. Employing learned rotations, whether under $R_{1,2}$ settings or $R_{1,2,3,4}$ settings, consistently enhances accuracy across various models and bit-width configurations. Notably, in the quantization of Mistral-7B, `SpinQuant`$_{had}$ secures an improvement exceeding 15.7 points over using random Hadamard rotations. Given that rotation optimization incurs a minimal time cost (only 30 minutes for smaller models and up to 3.5 hours for a 70B model) we advocate for the adoption of optimized rotations for precise quantization of LLMs.

### 4.3.2 COMPATIBILITY WITH GPTQ

In the context where both weights and activations are quantized, we observed that the learned rotations tend to adapt effectively to both weight and activation quantization. Given that GPTQ significantly helps mitigate the errors due to weight quantization, but leaves activation quantization untouched, we elect to optimize the rotation matrices with respect to a network where only activations are quantized. This approach allows the rotation to more efficiently manage the activation quantization error while leaving the weight quantization error to be addressed by GPTQ. As shown in Table 3, this modification resulted in superior performance in both W4A4 and W4A4KV4 settings in the LLaMA-2 7B model, which is the configuration we choose to utilize throughout the rest of this paper.

### 4.3.3 ROTATION TYPE

In Table 4, we evaluate the impact of random orthogonal floating-point rotation matrices and random Hadamard matrices on quantization accuracy, utilizing round-to-nearest quantization for our analysis. Prior to optimization, the Hadamard matrices yield a better-quantized network performance compared to floating-point rotation matrices. However, after optimization, the initial choice of rotation, whether floating-point or Hadamard, becomes less significant. This is likely due to the loss-aware rotation optimization's ability to locate an optimal local minima that effectively minimizes quantization error, thereby enhancing robustness to varying types of rotation initialization.

### 4.3.4 COMPARISON WITH QUAROT

Compared to QuaRot (Ashkboos et al., 2023b), which exhibits significant accuracy variances in quantized networks—experiencing drops of 28.1 and 33.2 points when quantizing a 70B model with round-to-nearest methods to W4A4 and W4A4KV4—this degradation stems from inherent noise in using random rotation matrix that introduce high variance and compromise robustness. In contrast, `SpinQuant`$_{had}$ consistently maintains high accuracy across various configurations, achieving improvements of 2.0 to 28.6 points over QuaRot (Table 5), while utilizing fewer online Hadamard matrices (two per block in `SpinQuant`$_{had}$ versus four per block in QuaRot).

Furthermore, the integration of $R_2$ in `SpinQuant` effectively reduces in-block outliers, thereby enabling `SpinQuant`$_{no\ had}$ to deliver optimal performance in W4A8 settings. `SpinQuant`$_{no\ had}$ can be achieved by simply substituting the model weights with rotated weights, making it a more

Table 4: Floating-point(FP) rotation vs Hadamard rotation on a LLaMA-2 7B model.

| #Bits (W-A-KV) | Task | No *Cayley* + RTN | | *Cayley* + RTN | |
| --- | --- | --- | --- | --- | --- |
| | | FP | Hadamard | FP init. | Hadamard init. |
| 4-16-16 | 0-shot[8] Avg.(↑) | 62.5 ±0.8 | 62.4 ±1.0 | 64.9 ±0.4 | 64.6 ±0.3 |
| | Wiki(↓) | 6.7 ±0.12 | 6.9 ±0.45 | 5.5 ±0.01 | 5.5 ±0.01 |
| 4-4-16 | 0-shot[8] Avg.(↑) | 49.4 ±2.8 | 59.0 ±1.0 | 61.6 ±0.4 | 61.8 ±0.4 |
| | Wiki(↓) | 15.9 ±4.04 | 8.2 ±0.73 | 6.2 ±0.06 | 6.1 ±0.03 |
| 4-4-4 | 0-shot[8] Avg.(↑) | 48.3 ±2.7 | 58.7 ±1.0 | 61.5 ±0.8 | 61.5 ±0.3 |
| | Wiki(↓) | 18.2 ±4.35 | 8.2 ±0.36 | 6.3 ±0.08 | 6.2 ±0.03 |

Table 5: Comparison with QuaRot (Ashkboos et al., 2023b).

| | LLaMA-3 8B (FP: 69.6, 6.1) | | | | LLaMA-3 70B (FP: 74.5, 2.8) | | | |
| --- | --- | --- | --- | --- | --- | --- | --- | --- |
| | 4-4-16 | | 4-4-4 | | 4-4-16 | | 4-4-4 | |
| | 0-shot[8] Avg.(↑) | Wiki (↓) | 0-shot[8] Avg.(↑) | Wiki (↓) | 0-shot[8] Avg.(↑) | Wiki (↓) | 0-shot[8] Avg.(↑) | Wiki (↓) |
| QuaRot+RTN | 59.5 | 10.4 | 58.6 | 10.9 | 41.5 | 91.2 | 41.3 | 92.4 |
| SpinQuant$_{had}$+RTN | **64.6** | **7.7** | **64.1** | **7.8** | **70.1** | **4.1** | **70.1** | **4.1** |
| QuaRot+GPTQ | 63.8 | 7.9 | 63.3 | 8.0 | 65.4 | 20.4 | 65.1 | 20.2 |
| SpinQuant$_{had}$+GPTQ | **65.8** | **7.1** | **65.5** | **7.3** | **69.5** | **5.5** | **69.3** | **5.5** |

straightforward and efficient approach compared to QuaRot, which requires modifying the model architecture and special kernel support.

## 4.4 ILLUSTRATIVE ANALYSIS OF THE ROTATION EFFICACY

The rationale behind rotating network weights and activations can be elucidated through a straightforward example. Consider an activation ($X$) represented as a 2D vector, where one entry $x_1$ consistently receives higher magnitude activations than $x_2$ (as depicted in Figure 6(a)). Quantizing these components together typically results in a quantization range dominated by $x_1$, thereby compromising the precision for $x_2$.

From an information entropy standpoint, expanding each axis to fully utilize the available quantization range maximizes the representational capacity of each axis. Thus, matrix rotation emerges as an intuitive solution. In a 2D scenario, rotating the axis by 45° equalizes the value representation range across axes (illustrated in Figure 6(b)). Assuming the network as a black box without knowledge of the exact activation distribution, uniformly rotating all axes by the maximal degree (45° in 2D) can optimize distribution evenness across each axis, partially explaining why Hadamard rotation often outperforms random rotation matrices.

Taking this further, if the activation distribution is known, treating the network as a white box during quantization allows for the identification of more optimal rotations than Hadamard. For instance, in a 3D scenario depicted in Figure 6(c-d), where $x_1$'s magnitude is four times that of $x_2$ and $x_3$, rotating the distribution by 45° along $x_3$ and $x_2$ redistributes the maximum values from $[2, 0.5, 0.5]$ to $[1, 1, 1.414]$. However, even more optimal rotation strategies may exist, and learning the rotation can help pinpoint the most effective rotation for a given distribution.

This opens up intriguing research avenues, such as determining if, given an activation distribution with known outlier axes and magnitudes, a closed-form solution for the optimal rotation matrix that evenly distributes magnitude across different axes can be derived. Additionally, it raises the question

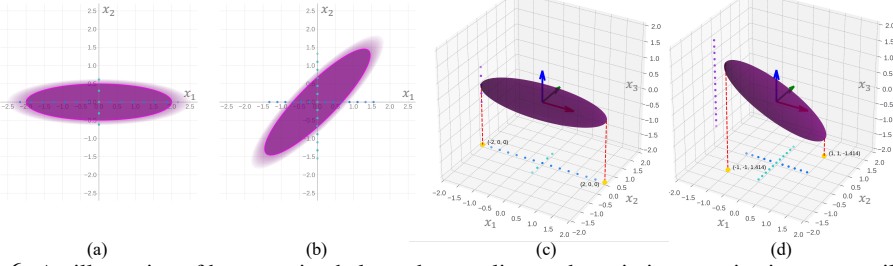

Figure 6: An illustration of how rotation helps reduce outliers and maximize quantization range utilization.

Table 6: Real-time end-to-end speed measurement of LLaMA-3 8B on MacBook M1 Pro CPU.

| Method | #Bits$_{(W\text{-}A)}$ | Decoding speed |
|---|---|---|
| FloatingPoint | 16-16 | 177.15 ms/token |
| SpinQuant$_{no\ had}$ | 4-8 | 58.88 ms/token |
| SpinQuant$_{had}$ | 4-8 | 63.90 ms/token |

of whether this theoretically calculated rotation yields the best quantization performance. We leave this question to future research.

## 4.5 SPEED MEASUREMENT

We conduct an end-to-end speed measurement of the LLaMA-3 8B model with W16A16 and W4A8 configurations on a MacBook M1 Pro CPU (OS version 14.5). The results in Table 6 demonstrate that 4-bit quantization yields a $\sim 3\times$ speedup compared to the 16-bit model. Comparing SpinQuant$_{had}$ to SpinQuant$_{no\ had}$, online Hadamard processing introduced a modest 8% increase in latency. Therefore, it is a trade-off between using SpinQuant$_{no\ had}$ without online Hadamard for its simpleness or using SpinQuant$_{had}$ with online Hadamard rotations for higher accuracy in lower-bit activation quantization. Detailed GPU latency results are provided in the Appendix.

## 5 RELATED WORK

**Quantization** Neural network quantization has been demonstrated as an effective tool for model size compression and storage reduction (Nagel et al., 2020; Krishnamoorthi, 2018; Nagel et al., 2019; Li et al., 2021). However, in large language models (LLMs), quantization presents unique challenges due to the presence of numerous outliers. These outliers dominate the quantization range, leaving only a few effective bits for the majority of values. Various strategies have been proposed to address the difficulties in LLM quantization. These include separating outliers and using mixed precision (Dettmers et al., 2022; van Baalen et al., 2023; Kim et al., 2023; Huang et al., 2024; Egiazarian et al., 2024), employing Hessian-based methods to mitigate quantization difficulty (Frantar et al., 2022), trading outliers between weights and activations (Xiao et al., 2022; Lin et al., 2023; Liu et al., 2023b) utilizing weight equalization (Nagel et al., 2019), outlier suppression (Wei et al., 2022; 2023), channel reassembly (Liu et al., 2023a) and even suggesting architectural modifications to handle outliers during pre-training(Yelysei Bondarenko, 2023). Recently two QuIP papers (Chee et al., 2024; Tseng et al., 2024) introduce the incoherence processing using random rotation matrices and applying vector quantization on the weights for compression. This does introduce extra overhead and imposes some constraints on the devices the LLM is deployed to in the availability of vector quantization kernels.

**Optimization in orthonormal space** The optimization of rotation matrices is carried out within the *Stiefel Manifold* (James, 1976), which encompasses all orthonormal matrices. Optimization while staying on this manifold can be done by *e.g.*, parameterizing a skew-symmetric matrix and applying the *Cayley* transformation on top of it (Nishimori and Akaho, 2005), or using a matrix exponential (Absil and Malick, 2012; Lezcano-Casado and Martinez-Rubio, 2019). However, these methods rely on expensive inverse or matrix-exponential functions that are applied every iteration. Instead, we follow the more efficient method named *Cayley SGD* (Li et al., 2020), which can be applied to optimize a rotation matrix $R$ for arbitrary loss functions efficiently. *Cayley SGD* relies on an iterative approximation of the *Cayley* Transform that is conducted solely with matrix multiplications.

## 6 CONCLUSIONS

In this paper, we present SpinQuant, a novel quantization technique that utilizes learned rotation to effectively bridge the performance gap between full precision and 4-bit weight, activation, and kv-cache quantization. At its core, SpinQuant leverages the rotation invariance property of LLM models to insert rotation matrices that diminish outliers in the weights and intermediate activations while maintaining the network's full-precision output numerically identical. Additionally, SpinQuant incorporates *Cayley SGD* for optimizing rotation matrices, resulting in improved and robust quantization outcomes. Importantly, SpinQuant is compatible with more advanced weight quantization techniques (*e.g.*, GPTQ) and demonstrates state-of-the-art performance.

## ACKNOWLEDGMENT

We extend our gratitude to Scott Wolchok and Chen Lai for their crucial contributions to latency measurement on the MacBook M1 Pro CPU, and to Sijia Chen, Geonhwa Jeong and Jiecao Yu for their expert support in GPU latency assessment.

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

## A  APPENDIX / SUPPLEMENTAL MATERIAL

### A.1  COMPLETE RESULTS OF MAIN RESULT TABLE

In Tables 7, 8 and 9, we show the complete results of Table 1. We compare the accuracy on eight zero-shot commonsense reasoning tasks including ARC-easy, ARC-challenge (Clark et al., 2018), BoolQ (Clark et al., 2019), PIQA (Bisk et al., 2020), SIQA (Sap et al., 2019), HellaSwag (Zellers et al., 2019), OBQA (Mihaylov et al., 2018), and WinoGrande (Sakaguchi et al., 2021) as well as the perplexity score on WikiText2 testset (Merity et al., 2016). We compare our results with previous works including SmoothQuant(Xiao et al., 2022), LLM-QAT(Liu et al., 2023c), GPTQ (Frantar et al., 2022), OmniQuant (Shao et al., 2023), QuIP# (Tseng et al., 2024).

### A.2  RESULTS ON 3-BIT WEIGHT QUANTIZATION

We present the 3-bit weight and 8-bit activation quantization results across seven models in Table 10. Our method, `SpinQuant`, successfully reduces the gap to the full-precision network from the previous $9.0 - 28.0$ points to $1.2 - 5.3$ points, demonstrating its effectiveness for low-bit quantization.

### A.3  *Cayley* OPTIMIZATION CHOICE

In Table 11, we evaluate the impact of varying the number of samples and iterations used in *Cayley* optimization. Given the limited trainable parameters in the rotation matrix and its constraint optimization nature, minimal calibration data and iterations are sufficient to optimize the rotation for better quantization. The findings indicate that rotation optimization is resilient to modifications in the number of samples. Even though we used 800 samples in our experiments, reducing this to 128 samples does not lead to a significant change in the perplexity. Furthermore, we examined the optimal number of iterations and found that the wiki perplexity ceases to decrease and stabilizes at 100 iterations. Consequently, we chose to use 100 iterations in all our experiments.

### A.4  QUANTIZATION CHOICE

We conduct an ablation study on symmetric vs asymmetric quantization and whether to clip the min-max ranges or not during activation and KV-cache quantization. The results in Table 12 show that for both activation quantization and KV-cache quantization, asymmetric quantization outperforms symmetric quantization. In the clip settings, we set the activation clipping ratio to 0.9 and the KV-cache clipping ratio to 0.95 as suggested in the previous works (Zhao et al., 2023). However, the results show that clipping the range or not does not impact the final result significantly. Therefore we opt for no clipping, *i.e.*, using the min-max quantization for activation and KV cache quantization across our experiments due to its simplicity.

### A.5  CALIBRATION DATA CHOICE

To assess the robustness of SpinQuant with respect to calibration data used in rotation optimization we use C4 dataset (Raffel et al., 2020) as calibration data and performe experiments on the LLaMA-2 7B model. The results in Table 13 reflect that using C4 datasets yields consistent results with utilizing the Wiki dataset, showing that SpinQuant is robust to calibration data choice.

### A.6  LATENCY MEASUREMENT ON GPU

In light of the available Tensor cores in NVIDIA's Hopper (H100) architecture, we provide the whole network end-to-end speed test result of W-fp8-A-fp8 quantization on H100 GPU, both with and without Hadamard transformations. Specifically, we utilize FP8 GEMM from the FBGEMM repo [3], which incorporates dequantization via epilogue fusion. We also leverage the Tensor Core-based Hadamard transform kernel [4] to minimize the overhead of the online Hadamard transform. The end-

---

[3] https://github.com/pytorch/FBGEMM/blob/main/fbgemm_gpu/experimental/gemm/triton_gemm/fp8_gemm.py

[4] https://github.com/pytorch-labs/applied-ai/blob/main/kernels/cuda/inference/hadamard_transform/hadamard_transform.cpp

Table 7: Complete comparison of the perplexity score on WikiText2 and averaged accuracy on Zero-shot Common Sense Reasoning tasks on **LLaMA-2**.

| Model | #Bits W-A-KV | Method | ARC-e (↑) | ARC-c (↑) | BoolQ (↑) | PIQA (↑) | SIQA (↑) | HellaS. (↑) | OBQA (↑) | WinoG. (↑) | Avg. (↑) | Wiki2 (↓) |
|---|---|---|---|---|---|---|---|---|---|---|---|---|
| 7B | 16-16-16 | Full Precision | 75.0 | 50.8 | 77.3 | 78.9 | 48.5 | 76.0 | 59.3 | 69.5 | 66.9 | 5.5 |
| | 4-8-16 | RTN | 70.9 | 44.3 | 73.5 | 76.8 | 46.0 | 70.3 | 51.8 | 65.9 | 62.4 | 7.9 |
| | | SmoothQuant | 65.8 | 41.7 | 67.3 | 75.6 | 44.5 | 67.1 | 45.8 | 63.5 | 58.9 | 7.5 |
| | | LLM-QAT | 73.6 | 49.0 | 72.4 | 78.2 | 47.8 | 74.0 | 56.1 | 67.7 | 64.8 | 11.4 |
| | | AWQ (w4) | – | – | – | – | – | – | – | – | – | 6.2 |
| | | OmniQuant (w4) | – | – | – | – | – | – | – | – | – | 5.7 |
| | | QuIP# (w4) | – | – | – | – | – | – | – | – | – | 5.6 |
| | | GPTQ | 73.7 | 47.5 | 74.8 | 77.7 | 46.4 | 74.1 | 55.7 | 69.3 | 64.9 | 20.2 |
| | | SpinQuant$_{no\ had}$ | 73.6 | 49.4 | 76.0 | 79.0 | 47.8 | 75.0 | 56.1 | 68.8 | 65.7 | 5.8 |
| | | SpinQuant$_{had}$ | 74.0 | 50.1 | 74.4 | 78.9 | 47.6 | 74.8 | 56.7 | 68.9 | 65.7 | 5.7 |
| | 4-8-8 | RTN | 71.1 | 44.3 | 73.2 | 76.8 | 45.8 | 70.3 | 52.3 | 65.8 | 62.5 | 7.9 |
| | | SmoothQuant | 65.8 | 40.8 | 66.4 | 76.3 | 43.7 | 66.9 | 46.0 | 64.5 | 58.8 | 7.5 |
| | | LLM-QAT | 73.5 | 48.3 | 72.4 | 78.1 | 47.4 | 74.0 | 55.3 | 68.0 | 64.6 | 11.4 |
| | | GPTQ | 73.7 | 48.0 | 74.2 | 78.1 | 46.6 | 73.9 | 55.1 | 68.5 | 64.8 | 20.2 |
| | | SpinQuant$_{no\ had}$ | 75.1 | 49.8 | 74.7 | 78.2 | 47.8 | 75.0 | 57.6 | 67.7 | 65.8 | 5.8 |
| | | SpinQuant$_{had}$ | 73.4 | 49.6 | 76.0 | 78.4 | 47.7 | 74.6 | 56.2 | 70.3 | 65.8 | 5.7 |
| | 4-4-16 | RTN | 26.6 | 22.1 | 44.3 | 50.9 | 38.9 | 26.2 | 26.6 | 49.4 | 35.6 | 2,167.2 |
| | | SmoothQuant | 37.8 | 27.1 | 51.9 | 59.4 | 40.2 | 34.3 | 31.6 | 52.4 | 41.8 | 254.5 |
| | | LLM-QAT | 46.2 | 32.4 | 61.8 | 62.0 | 41.3 | 47.6 | 36.1 | 54.7 | 47.8 | 12.9 |
| | | GPTQ | 27.6 | 24.9 | 47.4 | 50.7 | 38.6 | 26.9 | 28.3 | 49.9 | 36.8 | 8,949.0 |
| | | SpinQuant$_{no\ had}$ | 61.0 | 39.4 | 66.0 | 72.6 | 44.5 | 66.1 | 45.1 | 61.6 | 57.0 | 9.2 |
| | | SpinQuant$_{had}$ | 72.1 | 47.5 | 74.4 | 77.0 | 47.3 | 73.2 | 54.4 | 66.9 | 64.1 | 5.9 |
| | 4-4-4 | RTN | 27.1 | 24.4 | 44.8 | 51.4 | 39.4 | 26.7 | 33.0 | 50.0 | 37.1 | 2,382.5 |
| | | SmoothQuant | 31.4 | 24.8 | 51.4 | 54.1 | 39.4 | 29.1 | 31.9 | 50.0 | 39.0 | 698.7 |
| | | LLM-QAT | 42.0 | 27.7 | 59.5 | 58.9 | 41.0 | 43.1 | 33.5 | 53.3 | 44.9 | 14.9 |
| | | GPTQ | 27.6 | 23.6 | 47.8 | 51.0 | 38.7 | 27.0 | 28.5 | 50.3 | 36.8 | 9,253.1 |
| | | SpinQuant$_{no\ had}$ | 61.8 | 39.1 | 64.8 | 71.6 | 44.5 | 65.0 | 41.4 | 60.0 | 56.0 | 9.2 |
| | | SpinQuant$_{had}$ | 72.6 | 47.5 | 73.9 | 77.0 | 47.2 | 73.0 | 54.1 | 66.9 | 64.0 | 5.9 |
| 13B | 16-16-16 | Full Precision | 75.3 | 51.4 | 79.8 | 80.4 | 50.5 | 79.8 | 56.8 | 72.5 | 68.3 | 5.0 |
| | 4-8-16 | RTN | 63.1 | 39.9 | 68.7 | 74.0 | 46.2 | 59.7 | 45.5 | 61.5 | 57.3 | 6.7 |
| | | SmoothQuant | 71.7 | 46.3 | 72.0 | 78.2 | 47.3 | 72.8 | 51.2 | 69.2 | 63.6 | 6.1 |
| | | LLM-QAT | 75.3 | 49.7 | 79.0 | 80.0 | 50.3 | 77.4 | 56.3 | 71.6 | 67.5 | 14.5 |
| | | AWQ (w4) | – | – | – | – | – | – | – | – | – | 5.1 |
| | | OmniQuant (w4) | – | – | – | – | – | – | – | – | – | 5.0 |
| | | QuIP# (w4) | – | – | – | – | – | – | – | – | – | 5.0 |
| | | GPTQ | 74.2 | 49.2 | 75.3 | 78.4 | 48.8 | 74.1 | 53.4 | 68.4 | 65.2 | 5.9 |
| | | SpinQuant$_{no\ had}$ | 76.5 | 52.0 | 81.5 | 80.0 | 49.9 | 78.8 | 54.8 | 72.4 | 68.2 | 5.1 |
| | | SpinQuant$_{had}$ | 76.2 | 50.6 | 80.1 | 80.1 | 49.8 | 78.5 | 58.0 | 71.7 | 68.1 | 5.0 |
| | 4-8-8 | RTN | 63.2 | 40.3 | 69.0 | 74.3 | 46.1 | 59.5 | 46.2 | 61.9 | 57.6 | 6.7 |
| | | SmoothQuant | 73.3 | 45.3 | 71.9 | 78.8 | 47.6 | 72.7 | 49.6 | 67.7 | 63.4 | 6.1 |
| | | LLM-QAT | 75.0 | 48.8 | 79.2 | 80.3 | 50.7 | 77.7 | 56.1 | 72.3 | 67.5 | 14.2 |
| | | GPTQ | 74.1 | 48.8 | 75.1 | 78.1 | 48.8 | 74.1 | 53.6 | 69.5 | 65.3 | 5.9 |
| | | SpinQuant$_{no\ had}$ | 76.8 | 52.1 | 80.8 | 80.5 | 49.9 | 78.6 | 55.8 | 70.6 | 68.1 | 5.1 |
| | | SpinQuant$_{had}$ | 76.7 | 51.2 | 80.4 | 80.5 | 49.4 | 78.6 | 57.4 | 71.5 | 68.2 | 5.1 |
| | 4-4-16 | RTN | 26.0 | 26.0 | 40.6 | 49.7 | 38.7 | 26.0 | 25.4 | 49.9 | 35.3 | 7,216.7 |
| | | SmoothQuant | 45.2 | 27.1 | 55.4 | 62.5 | 40.5 | 44.3 | 33.4 | 50.8 | 44.9 | 34.5 |
| | | LLM-QAT | 26.0 | 23.7 | 37.8 | 49.2 | 39.5 | 26.3 | 23.8 | 48.2 | 34.3 | 3,889.9 |
| | | GPTQ | 26.6 | 24.7 | 37.9 | 49.3 | 39.2 | 26.2 | 27.7 | 50.3 | 35.2 | 5,245.3 |
| | | SpinQuant$_{no\ had}$ | 68.5 | 43.0 | 72.1 | 75.4 | 48.5 | 71.2 | 51.0 | 64.6 | 61.8 | 7.2 |
| | | SpinQuant$_{had}$ | 75.9 | 50.8 | 78.1 | 79.5 | 49.4 | 77.5 | 55.2 | 70.8 | 67.2 | 5.2 |
| | 4-4-4 | RTN | 26.1 | 24.3 | 40.3 | 48.7 | 39.6 | 25.8 | 29.2 | 49.6 | 35.5 | 7,428.8 |
| | | SmoothQuant | 36.9 | 24.8 | 49.4 | 57.2 | 39.6 | 33.3 | 31.2 | 51.7 | 40.5 | 56.6 |
| | | LLM-QAT | 26.3 | 24.6 | 37.8 | 48.8 | 39.3 | 26.3 | 26.8 | 50.4 | 35.0 | 3,777.5 |
| | | GPTQ | 26.6 | 24.1 | 37.9 | 48.8 | 38.9 | 26.1 | 29.3 | 50.1 | 35.2 | 5,237.1 |
| | | SpinQuant$_{no\ had}$ | 67.1 | 39.7 | 72.5 | 74.7 | 47.4 | 71.1 | 47.8 | 65.3 | 60.7 | 7.1 |
| | | SpinQuant$_{had}$ | 75.7 | 50.5 | 79.3 | 79.5 | 49.1 | 77.1 | 53.8 | 69.9 | 66.9 | 5.3 |
| 70B | 16-16-16 | Full Precision | 80.2 | 60.5 | 85.1 | 82.8 | 50.8 | 84.3 | 59.0 | 80.6 | 72.9 | 3.3 |
| | 4-8-16 | RTN | 78.2 | 54.8 | 81.5 | 80.8 | 46.9 | 76.5 | 56.5 | 73.3 | 68.6 | 5.0 |
| | | SmoothQuant | 79.4 | 57.3 | 82.4 | 82.0 | 50.3 | 81.5 | 56.2 | 75.9 | 70.6 | 4.1 |
| | | OmniQuant (w4) | – | – | – | – | – | – | – | – | – | 3.5 |
| | | QuIP# (w4) | – | – | – | – | – | – | – | – | – | 3.4 |
| | | GPTQ | 80.2 | 59.5 | 82.4 | 82.6 | 50.3 | 82.1 | 58.3 | 77.9 | 71.7 | 4.3 |
| | | SpinQuant$_{no\ had}$ | 80.0 | 59.2 | 84.4 | 82.6 | 50.3 | 82.8 | 59.7 | 78.1 | 72.1 | 3.7 |
| | | SpinQuant$_{had}$ | 80.2 | 59.9 | 85.0 | 82.5 | 50.4 | 83.9 | 60.1 | 79.3 | 72.7 | 3.5 |
| | 4-8-8 | RTN | 78.3 | 53.9 | 81.4 | 81.4 | 47.3 | 76.7 | 56.0 | 72.6 | 68.4 | 5.0 |
| | | SmoothQuant | 80.0 | 57.8 | 81.6 | 81.6 | 48.9 | 81.5 | 56.6 | 75.8 | 70.5 | 4.1 |
| | | GPTQ | 79.6 | 60.3 | 82.4 | 82.2 | 49.9 | 82.2 | 58.5 | 77.3 | 71.6 | 4.3 |
| | | SpinQuant$_{no\ had}$ | 80.4 | 60.3 | 84.4 | 81.8 | 49.8 | 82.8 | 59.1 | 79.0 | 72.2 | 3.7 |
| | | SpinQuant$_{had}$ | 80.4 | 59.7 | 85.2 | 82.6 | 50.4 | 83.8 | 59.9 | 79.8 | 72.7 | 3.5 |
| | 4-4-16 | RTN | 26.0 | 23.2 | 43.5 | 48.9 | 37.0 | 26.0 | 25.6 | 50.5 | 35.1 | 2e5 |
| | | SmoothQuant | 69.5 | 71.7 | 29.0 | 66.6 | 73.1 | 45.1 | 67.4 | 39.4 | 57.7 | 57.1 |
| | | GPTQ | 25.3 | 25.8 | 45.7 | 50.1 | 36.4 | 25.8 | 24.6 | 50.0 | 35.5 | 2e6 |
| | | SpinQuant$_{no\ had}$ | 66.8 | 42.4 | 72.9 | 74.0 | 46.7 | 73.2 | 48.2 | 63.9 | 61.0 | 7.3 |
| | | SpinQuant$_{had}$ | 78.4 | 57.0 | 82.7 | 81.4 | 50.2 | 83.0 | 58.5 | 77.0 | 71.0 | 3.8 |
| | 4-4-4 | RTN | 25.5 | 24.5 | 43.2 | 50.2 | 36.7 | 26.6 | 24.2 | 49.3 | 35.0 | 2e5 |
| | | SmoothQuant | 68.1 | 31.9 | 65.8 | 72.0 | 43.5 | 64.2 | 38.2 | 63.1 | 55.9 | 10.5 |
| | | GPTQ | 26.1 | 25.2 | 45.7 | 49.5 | 36.8 | 26.0 | 25.4 | 50.2 | 35.6 | 1e6 |
| | | SpinQuant$_{no\ had}$ | 68.2 | 42.0 | 74.1 | 73.8 | 46.9 | 74.3 | 50.0 | 66.8 | 62.0 | 7.4 |
| | | SpinQuant$_{had}$ | 78.3 | 57.6 | 82.1 | 81.7 | 50.1 | 82.9 | 59.8 | 77.3 | 71.2 | 3.8 |

Table 8: Complete omparison of the perplexity score on WikiText2 and averaged accuracy on Zero-shot Common Sense Reasoning tasks on **LLaMA-3**.

| Model | #Bits W-A-KV | Method | ARC-e (↑) | ARC-c (↑) | BoolQ (↑) | PIQA (↑) | SIQA (↑) | HellaS. (↑) | OBQA (↑) | WinoG. (↑) | Avg. (↑) | Wiki2 (↓) |
|---|---|---|---|---|---|---|---|---|---|---|---|---|
| **1B** | 16-16-16 | Full Precision | 65.2 | 38.7 | 69.5 | 75.3 | 44.8 | 60.7 | 40.2 | 60.9 | 56.9 | 13.4 |
| | 4-8-16 | RTN | 62.4 | 39.7 | 66.3 | 72.1 | 44.6 | 56.6 | 42.8 | 58.6 | 55.4 | 20.7 |
| | | SmoothQuant | 47.6 | 30.7 | 59.6 | 64.9 | 41.7 | 47.6 | 31.5 | 52.9 | 47.1 | 108.2 |
| | | LLM-QAT | 59.6 | 37.8 | 61.7 | 72.5 | 43.1 | 57.2 | 37.1 | 56.2 | 53.2 | 21.0 |
| | | GPTQ | 61.7 | 38.5 | 65.6 | 71.4 | 43.9 | 56.4 | 44.1 | 58.7 | 55.0 | 17.3 |
| | | SpinQuant$_{no\ had}$ | 60.9 | 39.5 | 65.9 | 73.2 | 46.1 | 57.7 | 44.3 | 60.3 | 56.0 | 15.3 |
| | | SpinQuant$_{had}$ | 60.8 | 39.8 | 66.5 | 73.9 | 44.7 | 59.0 | 46.9 | 60.8 | 56.5 | 14.4 |
| | 4-8-8 | RTN | 62.6 | 40.0 | 66.7 | 72.2 | 44.4 | 56.6 | 43.0 | 59.9 | 55.7 | 20.7 |
| | | SmoothQuant | 48.2 | 31.5 | 59.1 | 65.4 | 41.7 | 47.2 | 31.5 | 52.0 | 47.1 | 108.6 |
| | | LLM-QAT | 60.0 | 36.7 | 62.2 | 73.1 | 43.0 | 57.0 | 37.7 | 55.2 | 53.1 | 21.0 |
| | | GPTQ | 61.7 | 38.0 | 65.4 | 71.4 | 43.5 | 56.1 | 45.3 | 57.0 | 54.8 | 17.3 |
| | | SpinQuant$_{no\ had}$ | 61.2 | 40.6 | 64.9 | 72.7 | 45.1 | 58.3 | 43.2 | 59.9 | 55.7 | 15.3 |
| | | SpinQuant$_{had}$ | 59.2 | 37.3 | 66.4 | 73.6 | 44.9 | 59.1 | 46.3 | 59.2 | 55.8 | 14.3 |
| | 4-4-16 | RTN | 37.8 | 28.3 | 51.2 | 56.4 | 40.0 | 35.9 | 28.9 | 51.4 | 41.2 | 137.5 |
| | | SmoothQuant | 32.3 | 26.4 | 46.3 | 54.7 | 39.7 | 28.7 | 27.0 | 48.0 | 37.9 | 2,027.8 |
| | | LLM-QAT | 39.3 | 28.5 | 55.6 | 58.9 | 40.9 | 32.7 | 28.1 | 52.0 | 42.0 | 62.1 |
| | | GPTQ | 36.8 | 27.1 | 56.0 | 56.6 | 41.2 | 36.0 | 27.9 | 51.5 | 41.6 | 107.6 |
| | | SpinQuant$_{no\ had}$ | 44.8 | 29.7 | 61.2 | 59.7 | 40.2 | 41.0 | 32.4 | 49.8 | 44.8 | 48.4 |
| | | SpinQuant$_{had}$ | 59.3 | 37.1 | 64.6 | 69.9 | 44.4 | 55.4 | 41.2 | 56.0 | 53.5 | 15.3 |
| | 4-4-4 | RTN | 37.6 | 27.6 | 49.3 | 56.4 | 40.7 | 35.1 | 27.0 | 51.5 | 40.6 | 160.4 |
| | | SmoothQuant | 30.0 | 26.3 | 41.8 | 51.6 | 39.0 | 26.9 | 26.8 | 49.5 | 36.5 | 2,599.6 |
| | | LLM-QAT | 37.7 | 26.7 | 55.7 | 57.9 | 40.6 | 32.0 | 31.3 | 50.5 | 41.5 | 76.2 |
| | | GPTQ | 37.8 | 29.0 | 53.9 | 56.8 | 39.9 | 34.7 | 29.7 | 51.3 | 41.6 | 124.6 |
| | | SpinQuant$_{no\ had}$ | 45.4 | 30.7 | 59.2 | 60.7 | 41.4 | 40.8 | 32.6 | 51.4 | 45.3 | 47.7 |
| | | SpinQuant$_{had}$ | 59.4 | 39.4 | 64.4 | 68.9 | 43.4 | 54.6 | 41.4 | 55.9 | 53.4 | 15.9 |
| **3B** | 16-16-16 | Full Precision | 68.9 | 47.6 | 79.0 | 76.0 | 52.1 | 71.0 | 50.2 | 66.6 | 63.9 | 10.7 |
| | 4-8-16 | RTN | 60.2 | 42.6 | 70.9 | 72.6 | 49.7 | 66.2 | 43.6 | 62.7 | 58.6 | 29.0 |
| | | SmoothQuant | 59.8 | 40.7 | 59.2 | 73.8 | 46.9 | 65.5 | 40.7 | 58.5 | 55.6 | 288.5 |
| | | LLM-QAT | 64.7 | 46.1 | 74.1 | 75.4 | 49.3 | 69.9 | 45.3 | 61.4 | 60.8 | 41.1 |
| | | GPTQ | 60.8 | 41.4 | 71.9 | 73.6 | 47.7 | 65.9 | 43.4 | 65.0 | 58.7 | 25.2 |
| | | SpinQuant$_{no\ had}$ | 65.9 | 44.2 | 74.9 | 74.8 | 48.2 | 68.3 | 48.8 | 65.9 | 61.4 | 11.6 |
| | | SpinQuant$_{had}$ | 66.8 | 47.2 | 78.4 | 76.0 | 50.8 | 69.2 | 50.2 | 66.7 | 63.2 | 11.5 |
| | 4-8-8 | RTN | 60.2 | 41.3 | 71.3 | 73.1 | 49.6 | 66.2 | 42.6 | 63.0 | 58.4 | 28.8 |
| | | SmoothQuant | 59.5 | 39.3 | 57.9 | 73.5 | 46.6 | 65.3 | 41.9 | 60.1 | 55.5 | 281.3 |
| | | LLM-QAT | 65.2 | 45.1 | 74.5 | 76.1 | 49.1 | 69.6 | 43.9 | 60.7 | 60.5 | 39.3 |
| | | GPTQ | 61.0 | 42.0 | 72.5 | 72.7 | 47.9 | 66.3 | 43.4 | 63.6 | 58.7 | 24.1 |
| | | SpinQuant$_{no\ had}$ | 65.2 | 45.7 | 76.1 | 75.8 | 48.7 | 69.4 | 47.9 | 65.5 | 61.8 | 11.7 |
| | | SpinQuant$_{had}$ | 67.2 | 46.4 | 78.4 | 76.5 | 51.0 | 69.5 | 50.6 | 66.0 | 63.2 | 11.2 |
| | 4-4-16 | RTN | 41.0 | 29.8 | 43.8 | 57.3 | 41.8 | 41.4 | 31.1 | 50.9 | 42.1 | 741.9 |
| | | SmoothQuant | 43.6 | 30.5 | 52.8 | 58.0 | 40.4 | 37.7 | 33.1 | 52.9 | 43.6 | 372.3 |
| | | LLM-QAT | 47.3 | 30.9 | 60.8 | 63.8 | 42.4 | 43.2 | 35.9 | 51.1 | 46.9 | 37.6 |
| | | GPTQ | 42.0 | 30.0 | 44.8 | 60.1 | 41.2 | 44.7 | 34.0 | 50.5 | 43.4 | 264.4 |
| | | SpinQuant$_{no\ had}$ | 54.6 | 37.7 | 65.7 | 66.7 | 43.3 | 56.3 | 41.8 | 56.9 | 52.9 | 22.4 |
| | | SpinQuant$_{had}$ | 66.3 | 43.9 | 74.2 | 75.0 | 48.9 | 67.2 | 47.1 | 65.5 | 61.0 | 11.1 |
| | 4-4-4 | RTN | 38.4 | 26.9 | 41.3 | 58.3 | 39.9 | 40.0 | 32.2 | 52.9 | 41.2 | 799.7 |
| | | SmoothQuant | 36.4 | 26.2 | 50.4 | 55.8 | 39.0 | 30.3 | 30.2 | 52.2 | 40.0 | 553.2 |
| | | LLM-QAT | 44.4 | 29.7 | 61.5 | 62.0 | 42.3 | 41.2 | 33.8 | 52.4 | 45.9 | 42.0 |
| | | GPTQ | 38.2 | 25.1 | 42.0 | 56.6 | 41.5 | 44.1 | 31.1 | 50.5 | 41.1 | 352.6 |
| | | SpinQuant$_{no\ had}$ | 58.0 | 36.0 | 67.2 | 66.9 | 43.3 | 56.8 | 40.4 | 54.5 | 52.9 | 22.4 |
| | | SpinQuant$_{had}$ | 66.0 | 43.2 | 76.4 | 74.6 | 47.0 | 67.7 | 45.1 | 64.2 | 60.5 | 11.4 |
| **8B** | 16-16-16 | Full Precision | 77.6 | 57.7 | 83.3 | 80.7 | 48.7 | 79.6 | 55.8 | 73.7 | 69.6 | 6.1 |
| | 4-8-16 | RTN | 73.2 | 48.1 | 76.3 | 77.1 | 46.6 | 75.5 | 54.3 | 72.5 | 65.5 | 8.2 |
| | | SmoothQuant | 67.5 | 41.0 | 71.9 | 74.9 | 46.6 | 70.8 | 45.8 | 69.1 | 61.0 | 10.7 |
| | | LLM-QAT | 77.6 | 50.6 | 81.2 | 79.0 | 47.5 | 76.0 | 53.5 | 72.4 | 67.2 | 7.7 |
| | | GPTQ | 71.5 | 46.8 | 76.1 | 76.6 | 47.9 | 73.9 | 52.1 | 70.7 | 64.5 | 7.2 |
| | | SpinQuant$_{no\ had}$ | 77.8 | 55.4 | 80.6 | 79.9 | 48.9 | 77.5 | 55.5 | 73.3 | 68.6 | 6.7 |
| | | SpinQuant$_{had}$ | 76.5 | 54.0 | 81.5 | 79.6 | 48.6 | 78.1 | 56.4 | 72.4 | 68.4 | 6.5 |
| | 4-8-8 | RTN | 73.7 | 49.1 | 76.5 | 77.1 | 46.7 | 75.5 | 50.8 | 73.4 | 65.3 | 8.2 |
| | | SmoothQuant | 66.6 | 41.8 | 73.2 | 74.1 | 45.9 | 71.1 | 48.2 | 66.5 | 60.9 | 10.7 |
| | | LLM-QAT | 77.2 | 50.6 | 81.5 | 79.3 | 47.7 | 76.3 | 52.0 | 70.6 | 66.9 | 7.6 |
| | | GPTQ | 71.5 | 46.9 | 76.6 | 76.2 | 48.5 | 73.7 | 52.1 | 71.0 | 64.6 | 7.2 |
| | | SpinQuant$_{no\ had}$ | 77.2 | 56.2 | 81.5 | 79.2 | 48.8 | 77.2 | 56.1 | 72.9 | 68.6 | 6.7 |
| | | SpinQuant$_{had}$ | 77.6 | 57.4 | 81.3 | 80.2 | 48.6 | 78.1 | 55.5 | 72.0 | 68.8 | 6.5 |
| | 4-4-16 | RTN | 42.7 | 29.5 | 54.0 | 57.8 | 39.9 | 41.2 | 36.9 | 49.4 | 43.9 | 241.6 |
| | | SmoothQuant | 36.3 | 26.3 | 50.6 | 54.1 | 40.3 | 31.4 | 30.6 | 52.9 | 40.3 | 867.5 |
| | | LLM-QAT | 44.1 | 29.7 | 58 | 61.5 | 42.1 | 39.9 | 33 | 51.3 | 44.9 | 42.9 |
| | | GPTQ | 39.7 | 27.6 | 40.8 | 58.5 | 41.7 | 31.9 | 32.0 | 53.1 | 40.6 | 187.9 |
| | | SpinQuant$_{no\ had}$ | 56.5 | 35.3 | 53.3 | 68.0 | 44.5 | 59.9 | 37.5 | 59.7 | 51.9 | 18.6 |
| | | SpinQuant$_{had}$ | 75 | 50.9 | 78.9 | 77.5 | 47.2 | 75.9 | 52.9 | 68.5 | 65.8 | 7.1 |
| | 4-4-4 | RTN | 39.5 | 27.5 | 54.6 | 57.7 | 41.4 | 39.4 | 32.6 | 51.9 | 43.1 | 260.9 |
| | | SmoothQuant | 33.5 | 25.1 | 49.6 | 53.1 | 40.3 | 28.8 | 29.6 | 49.6 | 38.7 | 1,530.50 |
| | | LLM-QAT | 40.5 | 26.6 | 52.7 | 59.9 | 42.3 | 37.5 | 33.6 | 52.7 | 43.2 | 52.5 |
| | | GPTQ | 40.6 | 26.5 | 40.9 | 58.0 | 41.5 | 31.9 | 33.0 | 51.8 | 40.5 | 195.8 |
| | | SpinQuant$_{no\ had}$ | 58.4 | 37.1 | 54.7 | 67.7 | 43.4 | 60.1 | 41.2 | 57.9 | 52.6 | 18.6 |
| | | SpinQuant$_{had}$ | 75.1 | 51.2 | 77.2 | 77.3 | 47.6 | 75.2 | 54.1 | 66.2 | 65.5 | 7.3 |

to-end speed test results of LLaMA-3 70B are detailed in Table 14. When implemented meticulously, SpinQuant with Hadamard rotation sees marginal difference in the latency compared to without Hadamard rotation.

Table 9: Complete comparison of the perplexity score on WikiText2 and averaged accuracy on Zero-shot Common Sense Reasoning tasks on **Mistral-7B-v0.3**.

| #Bits W-A-KV | Method | ARC-e (↑) | ARC-c (↑) | BoolQ (↑) | PIQA (↑) | SIQA (↑) | HellaS. (↑) | OBQA (↑) | WinoG. (↑) | Avg. (↑) | Wiki2 (↓) |
|---|---|---|---|---|---|---|---|---|---|---|---|
| 16-16-16 | Full Precision | 81.0 | 57.9 | 84.2 | 82.1 | 48.2 | 80.8 | 59.6 | 73.8 | 71.0 | 5.4 |
| 4-8-16 | RTN | 53.4 | 49.0 | 78.4 | 67.6 | 45.6 | 59.7 | 54.3 | 66.3 | 59.3 | 6.8 |
|  | GPTQ | 38.4 | 41.4 | 74.7 | 59.8 | 42.3 | 45.5 | 50.6 | 61.1 | 51.7 | 8.6 |
|  | SpinQuant$_{no\ had}$ | 75.4 | 55.7 | 81.9 | 80.3 | 48.2 | 78.1 | 57.8 | 72.7 | 68.8 | 5.7 |
|  | SpinQuant$_{had}$ | 78.9 | 55.9 | 82.7 | 81.9 | 48.5 | 80.0 | 58.4 | 72.7 | 69.9 | 5.5 |
| 4-8-8 | RTN | 52.9 | 48.7 | 78.5 | 67.3 | 45.5 | 59.4 | 52.7 | 66.4 | 58.9 | 6.7 |
|  | GPTQ | 38.7 | 40.6 | 74.8 | 58.9 | 42.5 | 45.8 | 51.0 | 61.3 | 51.7 | 8.6 |
|  | SpinQuant$_{no\ had}$ | 76.7 | 54.5 | 82.2 | 80.3 | 50.3 | 78.6 | 59.0 | 73.4 | 69.4 | 5.7 |
|  | SpinQuant$_{had}$ | 80.1 | 56.9 | 83.9 | 81.5 | 48.6 | 79.9 | 57.2 | 73.0 | 70.2 | 5.5 |
| 4-4-16 | RTN | 39.9 | 24.7 | 50.0 | 57.8 | 39.7 | 34.7 | 33.8 | 50.4 | 41.4 | 449.5 |
|  | GPTQ | 39.4 | 27.1 | 43.8 | 57.3 | 38.4 | 35.6 | 31.4 | 50.0 | 40.4 | 260.8 |
|  | SpinQuant$_{no\ had}$ | 55.2 | 34.6 | 67.9 | 70.8 | 41.9 | 50.8 | 44.7 | 56.0 | 52.7 | 13.4 |
|  | SpinQuant$_{had}$ | 76.5 | 53.3 | 80.7 | 80.7 | 48.2 | 78.6 | 57.8 | 71.2 | 68.4 | 5.7 |
| 4-4-4 | RTN | 39.9 | 26.7 | 51.2 | 58.1 | 40.3 | 34.4 | 28.7 | 51.7 | 41.4 | 443.5 |
|  | GPTQ | 40.4 | 28.5 | 43.6 | 57.4 | 39.2 | 35.2 | 33.8 | 52.1 | 41.3 | 249.9 |
|  | SpinQuant$_{no\ had}$ | 55.4 | 33.3 | 68.5 | 71.4 | 42.4 | 50.9 | 41.0 | 56.3 | 52.4 | 13.7 |
|  | SpinQuant$_{had}$ | 77.3 | 52.5 | 80.2 | 80.3 | 48.9 | 79.2 | 58.4 | 72.3 | 68.6 | 5.8 |

Table 10: 3-bit weight 8-bit activation quantization results on WikiText2 and Zero-shot Common Sense Reasoning tasks.

| #Bits W-A-KV | Method | ARC-e (↑) | ARC-c (↑) | BoolQ (↑) | PIQA (↑) | SIQA (↑) | HellaS. (↑) | OBQA (↑) | WinoG. (↑) | Avg. (↑) | Wiki2 (↓) |
|---|---|---|---|---|---|---|---|---|---|---|---|
| LLaMA-2 7B | *Full Precision* | 75.0 | 50.8 | 77.3 | 78.9 | 48.5 | 76.0 | 59.3 | 69.5 | 66.9 | 5.5 |
|  | RTN | 31.3 | 22.6 | 39.6 | 54.6 | 38.0 | 28.1 | 29.3 | 49.8 | 36.7 | 955.1 |
|  | SmoothQuant | 26.4 | 26.5 | 39.2 | 48.8 | 39.4 | 26.0 | 25.8 | 49.2 | 35.1 | 275,935.2 |
|  | LLM-QAT | 44.0 | 29.5 | 64.4 | 63.3 | 42.2 | 52.7 | 32.6 | 52.3 | 47.6 | 15.2 |
|  | GPTQ | 63.8 | 40.2 | 67.3 | 73.1 | 43.3 | 63.5 | 46.9 | 65.5 | 57.9 | 14.6 |
|  | SpinQuant$_{had}$ | 71.9 | 47.5 | 74.6 | 76.4 | 47.0 | 71.2 | 53.4 | 67.9 | 63.7 | 6.2 |
| LLaMA-2 13B | *Full Precision* | 75.3 | 51.4 | 79.8 | 80.4 | 50.5 | 79.8 | 56.8 | 72.5 | 68.3 | 5.0 |
|  | RTN | 30.4 | 24.6 | 48.8 | 53.8 | 39.8 | 29.0 | 25.4 | 49.5 | 37.6 | 167.8 |
|  | SmoothQuant | 26.1 | 25.5 | 37.8 | 49.0 | 39.4 | 26.1 | 26.4 | 49.5 | 35.0 | 8,979.3 |
|  | LLM-QAT | 27.5 | 20.7 | 40.1 | 51.1 | 38.2 | 26.4 | 27.9 | 50.7 | 35.3 | 256.6 |
|  | GPTQ | 56.5 | 34.5 | 63.3 | 68.9 | 44.2 | 46.0 | 39.8 | 56.3 | 51.2 | 10.8 |
|  | SpinQuant$_{had}$ | 75.9 | 52.4 | 76.6 | 78.4 | 49.3 | 74.6 | 56.2 | 70.6 | 66.7 | 5.4 |
| LLaMA-2 70B | *Full Precision* | 80.2 | 60.5 | 85.1 | 82.8 | 50.8 | 84.3 | 59.0 | 80.6 | 72.9 | 3.3 |
|  | RTN | 51.5 | 30.0 | 59.5 | 65.5 | 40.8 | 40.3 | 31.2 | 51.4 | 46.3 | 66.2 |
|  | SmoothQuant | 26.9 | 22.7 | 38.4 | 49.0 | 38.6 | 25.6 | 25.2 | 52.0 | 34.8 | 6,682.0 |
|  | GPTQ | 72.5 | 49.3 | 72.1 | 76.7 | 46.3 | 69.9 | 51.8 | 72.2 | 63.9 | 9.0 |
|  | SpinQuant$_{had}$ | 79.4 | 58.7 | 84.4 | 81.6 | 50.5 | 82.3 | 58.3 | 78.6 | 71.7 | 3.8 |
| LLaMA-3 1B | *Full Precision* | 65.2 | 38.7 | 69.5 | 75.3 | 44.8 | 60.7 | 40.2 | 60.9 | 56.9 | 13.4 |
|  | RTN | 32.6 | 28.0 | 54.8 | 55.7 | 39.1 | 34.2 | 29.7 | 47.8 | 40.2 | 2,097.6 |
|  | SmoothQuant | 28.8 | 24.0 | 40.4 | 51.6 | 37.8 | 25.9 | 28.2 | 48.0 | 35.6 | 58,367.5 |
|  | LLM-QAT | 47.0 | 30.4 | 60.3 | 62.8 | 41.6 | 39.9 | 33.6 | 51.8 | 45.9 | 46.9 |
|  | GPTQ | 41.5 | 30.4 | 61.4 | 62.3 | 39.9 | 41.7 | 33.0 | 50.6 | 45.1 | 90.8 |
|  | SpinQuant$_{had}$ | 58.8 | 36.4 | 63.7 | 68.7 | 44.2 | 51.5 | 38.1 | 56.5 | 52.2 | 17.2 |
| LLaMA-3 3B | *Full Precision* | 68.9 | 47.6 | 79.0 | 76.0 | 52.1 | 71.0 | 50.2 | 66.6 | 63.9 | 10.7 |
|  | RTN | 40.1 | 29.5 | 48.8 | 59.3 | 41.6 | 46.0 | 34.4 | 53.4 | 44.1 | 1,178.9 |
|  | SmoothQuant | 27.8 | 21.6 | 38.4 | 50.2 | 38.0 | 25.4 | 26.0 | 50.4 | 34.7 | 17,409.2 |
|  | LLM-QAT | 32.1 | 29.4 | 55.7 | 53.3 | 39.7 | 41.9 | 29.5 | 50.4 | 41.5 | 26.2 |
|  | GPTQ | 48.4 | 33.0 | 65.5 | 63.6 | 41.7 | 57.8 | 38.7 | 57.8 | 50.8 | 176.3 |
|  | SpinQuant$_{had}$ | 61.8 | 41.4 | 78.2 | 73.0 | 47.4 | 63.3 | 41.0 | 62.8 | 58.6 | 13.7 |
| LLaMA-3 8B | *Full Precision* | 77.6 | 57.7 | 83.3 | 80.7 | 48.7 | 79.6 | 55.8 | 73.7 | 69.6 | 6.1 |
|  | RTN | 40.9 | 25.3 | 62.3 | 58.8 | 39.7 | 35.1 | 31.4 | 54.1 | 43.5 | 196.2 |
|  | SmoothQuant | 27.4 | 24.9 | 38.3 | 50.9 | 37.9 | 25.7 | 29.8 | 49.8 | 35.6 | 179,664.5 |
|  | LLM-QAT | 35.9 | 28.0 | 54.3 | 58.5 | 39.8 | 31.7 | 27.7 | 50.9 | 40.8 | 14.9 |
|  | GPTQ | 50.8 | 34.5 | 65.6 | 64.0 | 42.4 | 55.1 | 37.3 | 61.5 | 51.4 | 9.4 |
|  | SpinQuant$_{had}$ | 74.5 | 50.3 | 79.6 | 77.2 | 46.8 | 74.5 | 50.6 | 70.9 | 65.5 | 7.5 |
| Mistral 7B | *Full Precision* | 81.0 | 57.9 | 84.2 | 82.1 | 48.2 | 80.8 | 59.6 | 73.8 | 71.0 | 5.4 |
|  | RTN | 28.2 | 28.1 | 62.2 | 53.1 | 38.7 | 28.0 | 35.9 | 48.3 | 40.3 | 167.1 |
|  | GPTQ | 31.9 | 32.7 | 63.8 | 54.8 | 40.0 | 31.0 | 36.9 | 52.2 | 42.9 | 29.3 |
|  | SpinQuant$_{had}$ | 77.7 | 54.1 | 82.2 | 79.9 | 47.7 | 77.5 | 59.4 | 73.8 | 69.0 | 5.8 |

Table 11: Ablation study on Number of training samples and iterations in *Cayley SGD* optimization, using LLaMA-2 7B.

| #Bits (W-A-KV) | Task | # Training sample | | # Training iterations | | | | |
|---|---|---|---|---|---|---|---|---|
|  |  | 128 | 800 | 10 | 25 | 50 | 100 | 200 |
| 4-4-4 | Wiki (↓) | 6.2 $_{\pm0.03}$ | 6.2 $_{\pm0.03}$ | 6.6 $_{\pm0.02}$ | 6.4 $_{\pm0.02}$ | 6.3 $_{\pm0.03}$ | 6.2 $_{\pm0.03}$ | 6.2 $_{\pm0.05}$ |

Table 12: Ablation of symmetric and asymmetric quantization and range clipping options on LLaMA-2 7B.

| #Bits | K asym | K clip | A asym | A clip | RTN | | GPTQ | |
|---|---|---|---|---|---|---|---|---|
| (W-A-KV) | | | | | Zero-shot Avg. ($\uparrow$) | Wiki ($\downarrow$) | Zero-shot Avg. ($\uparrow$) | Wiki ($\downarrow$) |
| 4-4-16 | – | – | ✗ | ✗ | 61.2 $_{\pm 0.6}$ | 6.3 | 63.3 $_{\pm 0.4}$ | 6.0 |
| 4-4-16 | – | – | ✓ | ✗ | 61.8 $_{\pm 0.4}$ | 6.1 | 64.0 $_{\pm 0.5}$ | 5.9 |
| 4-4-16 | – | – | ✓ | ✓ | 62.1 $_{\pm 0.6}$ | 6.0 | 64.0 $_{\pm 0.4}$ | 5.9 |
| 4-4-4 | ✗ | ✗ | ✓ | ✗ | 61.4 $_{\pm 0.5}$ | 6.2 | 63.7 $_{\pm 0.4}$ | 6.0 |
| 4-4-4 | ✓ | ✗ | ✓ | ✗ | 61.5 $_{\pm 0.6}$ | 6.2 | 63.7 $_{\pm 0.3}$ | 5.9 |
| 4-4-4 | ✓ | ✓ | ✓ | ✗ | 61.5 $_{\pm 0.3}$ | 6.2 | 63.7 $_{\pm 0.2}$ | 5.9 |

Table 13: Ablation study on calibration data choice using LLaMA-2 7B.

| Calibration Data | #Bits (W-A-KV) | ARC-e ($\uparrow$) | ARC-c ($\uparrow$) | BoolQ ($\uparrow$) | PIQA ($\uparrow$) | SIQA ($\uparrow$) | HellaS. ($\uparrow$) | OBQA ($\uparrow$) | WinoG. ($\uparrow$) | Avg. ($\uparrow$) | Wiki2 ($\downarrow$) |
|---|---|---|---|---|---|---|---|---|---|---|---|
| Wiki2 | 4-4-16 | 72.1 | 47.5 | 74.4 | 77.0 | 47.3 | 73.2 | 54.4 | 66.9 | 64.1 | 5.9 |
| Wiki2 | 4-4-4 | 72.6 | 47.5 | 73.9 | 77.0 | 47.2 | 73.0 | 54.1 | 66.9 | 64.0 | 5.9 |
| C4 | 4-4-16 | 72.5 | 47.3 | 74.8 | 77.6 | 47.7 | 73.2 | 55.4 | 66.2 | 64.3 | 5.9 |
| C4 | 4-4-4 | 72.5 | 47.9 | 74 | 78.4 | 46.7 | 73.1 | 55.5 | 66.4 | 64.3 | 6 |

## A.7 OPTIMIZATION TIME

In Table 15, we show the comparison of optimization time between GPTQ and SpinQuant. SpinQuant requires a scale of optimization time similar to that of GPTQ. The additional optimization time required by SpinQuant is worthwhile considering the substantial improvements it offers over GPTQ.

## A.8 ABLATION STUDY ON RTN VS GPTQ

SpinQuant is fully compatible with both GPTQ and naive RTN. To isolate the contributions of GPTQ and rotation to overall performance, we present results for SpinQuant combined with RTN in the W4A4KV16 quantization scenario in Table 16. Our analysis indicates that the primary accuracy gains are attributed to the incorporation of learned rotations, which enhances accuracy by $6.5 \sim 20.9$ percentage points over previous methods (including GPTQ). The subsequent integration of GPTQ further boosts performance by up to 2.3 percentage points.

## A.9 WEIGHT-ONLY QUANTIZATION

We also include a comparison of SpinQuant performance under weight-only quantization in Table 17. The weight-only results show that SpinQuant consistently achieves higher accuracy than AWQ and other previous work.

## A.10 FEW-SHOT RESULTS ON INSTRUCTION-FINETUNED MODELS

We further conduct experiments applying SpinQuant to instruction-finetuned LLaMA 3.2 1B and 3B models in Table 18. We present the results for few-shot learning scenarios. SpinQuant W4A8 quantized models demonstrate significant improvements in 5-shot accuracy on the MMLU benchmark and 1-shot rouge score on the TLDR9 summarization benchmark. It significantly closed the gap to the BF16 baseline.

# B ANALYSIS

## B.1 GRADIENT ANALYSIS

On the one hand, we have shown that the class of LLMs we are interested in are rotation invariant, *i.e.* the full-precision model output does not change regardless of what $R$ is. On the other hand, we are claiming that some $R$ are better than others for quantized LLM and that better $R$ can be learned with backpropagation on equation (2). To reconcile these seemingly conflicting claims, we inspect the gradient of the output of a single linear, $W$, and activations, $X$, which are both rotated and quantized:

$$\frac{\partial \sum_{ij} \left( Q(WR^{-1})Q(RX) \right)_{ij}}{\partial R_{mn}} = \sum_{ij} -(WR^{-1})_{im}(R^{-1}Q(RX))_{nj} + Q(WR^{-1})_{im}X_{nj} \quad (5)$$

Table 14: Real-time end-to-end speed measurement of LLaMA-3 70B on NVIDIA's Hopper (H100) GPU. TTFT (Time to First Token) and TTIT (Time to Iterative Token) are performance metrics to measure the pre-filling speed and decoding speed, respectively.

|  | SpinQuant without hadamard | | SpinQuant with hadamard | |
| --- | --- | --- | --- | --- |
|  | TTFT (ms) | TTIT (ms) | TTFT (ms) | TTIT (ms) |
| BS=1 T=4096 | 153.58 | 9.85 | 158.25 | 10.15 |
| BS=8 T=4096 | 1205.47 | 10.6 | 1243.48 | 10.94 |
| BS=32 T=4096 | 5008.25 | 13.83 | 5147.59 | 14.2 |

Table 15: Optimization time comparison between GPTQ and SpinQuant.

|  | llama3 1B | llama3 3B | llama3 8B | llama2 7B | llama2 13B | Mistral |
| --- | --- | --- | --- | --- | --- | --- |
| SpinQuant | 13 min | 18 min | 30 min | 25 min | 30 min | 16 min |
| GPTQ | 8 min | 13 min | 20 min | 18 min | 25 min | 12 min |

We see that equation (5):

- is non-zero in general, which validates our approach of using backpropagation to learn $R$

- reduces to 0 when quantization is not present, which validates the claim that it only makes sense to learn $R$ for quantized models

- demonstrates that two components move the gradient with respect to $R$ away from 0: 1) differences in quantized and unquantized rotated weights; 2) differences in quantized and unquantized rotated activations

## B.2 LOSS ANALYSIS

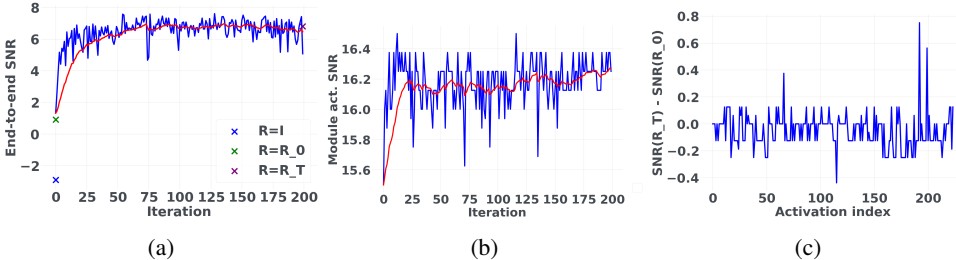

(a)                              (b)                              (c)

Figure 7: Training curves for LLaMA-2 7B with 4-bit weights and 4-bit activations in wiki2 train set. (a) End-to-end quantization SNR. $R_0$ and $R_T$ denote randomly initialized rotation and learned rotation after $T = 200$ iterations; (b) Activation quantization. SNR for layer 27 attention out projection; (c) Improvement in activation quantization SNR after optimization of $R$ for each layer.

While Sec. 4 shows that learning $R$ yields significant benefits on zero-shot reasoning tasks, in this section we shed some light on why our method is able to achieve accuracy gains. Intuitively, we expect the end-to-end signal to (quantization) noise ratio (SNR) to improve as a result of learning $R$. In other words, learning $R$ should bring the quantized model output closer to the floating point model output. As Table 19 shows, we observe an SNR improvement of 3.8 dB when introducing a random $R$ into LLaMA-2 7B with weights/activations quantized to 4 bits, and then an additional $5.9$dB improvement after learning $R$, all measured on the WikiText2 (Merity et al., 2016) test set. Figure 7a shows that the batch-level training set SNR during $R$ training progressively improves as expected, as well as the layer-level SNR for a particular layer in Figure 7b. Digging a bit deeper, Figure 7c shows the layer-level SNR improvement for each layer as a result of training $R$. We see that, perhaps counter-intuitively, layer-level SNR improves significantly for a few layers, but does not change much for most layers, and even gets worse for one of the layers. We hypothesize that: 1) certain layers have a disproportionate impact on model output or have a disproportionately low quantization SNR without rotation; 2) The process of optimizing $R$ rotates the residual stream basis such as to prioritize improving the SNR of such layers, possibly at the cost of hurting less important layers.

Table 16: Ablation study on SpinQuant combined with RTN or GPTQ in the W4A4KV16 quantization scenario.

| Method | LLaMA-3 8B Zero-shot | Wiki | LLaMA-2 7B Zero-shot | Wiki | LLaMA-2 13B Zero-shot | Wiki | LLaMA-2 70B Zero-shot | Wiki |
|---|---|---|---|---|---|---|---|---|
| Full-precision | 69.6 | 6.1 | 66.9 | 5.5 | 68.3 | 5.0 | 72.9 | 3.3 |
| RTN | 43.9 | 2e2 | 35.6 | 2e3 | 35.3 | 7e3 | 35.1 | 2e5 |
| SmoothQuant | 40.3 | 9e2 | 41.8 | 3e2 | 44.9 | 34.5 | 57.7 | 57.1 |
| LLM-QAT | 44.9 | 42.9 | 47.8 | 12.9 | 34.3 | 4e3 | – | – |
| GPTQ | 40.6 | 2e2 | 36.8 | 9e3 | 35.2 | 5e3 | 35.5 | 2e6 |
| SpinQuant$_{had}$ (RTN) | 64.6 | 7.7 | 61.8 | 6.1 | 65.8 | 5.4 | 71.1 | 3.9 |
| SpinQuant$_{had}$ (GPTQ) | **65.8** | **7.1** | **64.1** | **5.9** | **67.2** | **5.2** | **71.0** | **3.8** |

Table 17: A comparison of SpinQuant performance under 4-bit weight-only quantization.

| Method | LLaMA-2 7B Zero-shot Avg. (↑) | Wiki ppl(↓) | LLaMA-2 13B Zero-shot Avg.(↑) | Wiki ppl(↓) | LLaMA-2 70B Zero-shot Avg.(↑) | Wiki ppl(↓) |
|---|---|---|---|---|---|---|
| Full-precision | 66.9 | 5.5 | 68.3 | 5.0 | 72.9 | 3.3 |
| RTN | 63.6 | 7.2 | 57.9 | 6.4 | 69.2 | 4.6 |
| SmoothQuant | 59.1 | 7.5 | 63.3 | 6.1 | 70.2 | 4.1 |
| GPTQ | 64.5 | 11.3 | 64.7 | 5.6 | 71.9 | 3.9 |
| AWQ | – | 6.2 | – | 5.1 | – | – |
| SpinQuant$_{had}$ | **65.9** | **5.6** | **68.5** | **5.0** | **72.6** | **3.5** |

Table 18: Results of applying SpinQuant to instruction-finetuned LLaMA 3.2 1B and 3B models.

| | llama3.2 1B BF16 | Vanilla RTN | SpinQuant | llama3.2 3B BF16 | Vanilla RTN | SpinQuant |
|---|---|---|---|---|---|---|
| MMLU (5-shot) | 49.3 | 43.4 | 47.3 | 63.4 | 60.5 | 62.0 |
| TLDR9+ (test, 1-shot rougeL) | 16.8 | 14.9 | 16.7 | 19.0 | 19.1 | 19.2 |

Table 19: Average end-to-end signal to quantization noise ratio (dB) for LLaMA-2 7B with weights and activations quantized to 4 bits on wiki2 test set

| $R = I$ | Randomly initialized $R$ | Learned $R$ |
|---|---|---|
| -2.9 | 0.9 | 6.8 |

## C DISTRIBUTION VISUALIZATIONS BEFORE AND AFTER ROTATION

We present visualizations of the activation distributions before and after rotation in Figures 8 and 9, respectively. Similarly, the weight distributions before and after rotation are depicted in Figures 10 and 11. Overall, after rotation, the extreme values are attenuated, and the distribution exhibits no noteworthy outliers across the token dimension. Additionally, we make an interesting observation: in several activation layers, the first token displays substantial values in multiple channels. After rotation, this outlier is distributed across all channels of the first token. Although per-token activation quantization can readily manage this distribution, investigating the source of these outliers and reducing them prior to applying SpinQuant might further enhance quantization accuracy, which could be a potential future research direction.

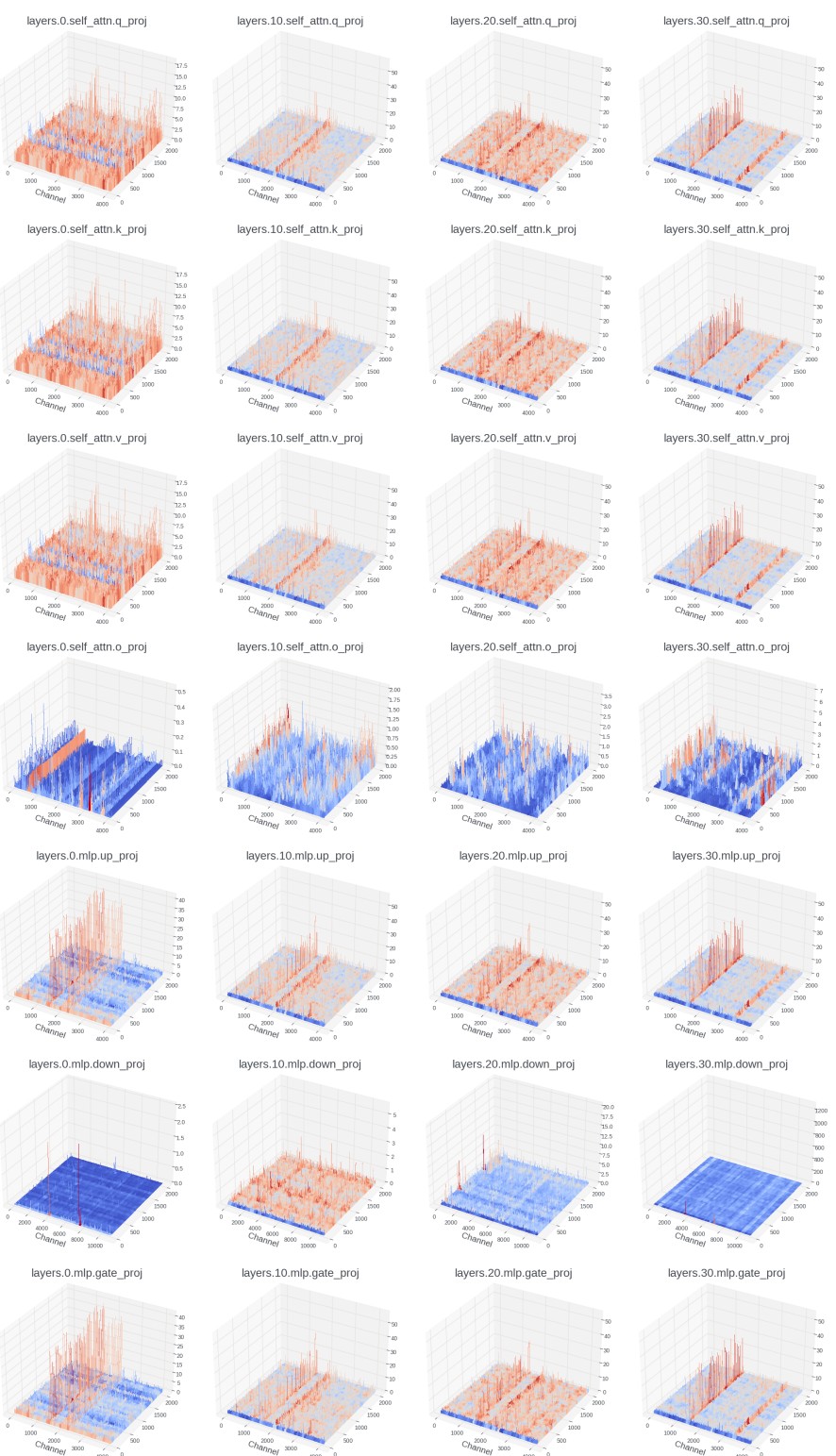

Figure 8: Magnitude of the input activations of a linear layer in $\{1^{st}, 11^{th}, 21^{st},$ and $31^{st}\}$ blocks in LLaMA-2 7B model **before rotation**.

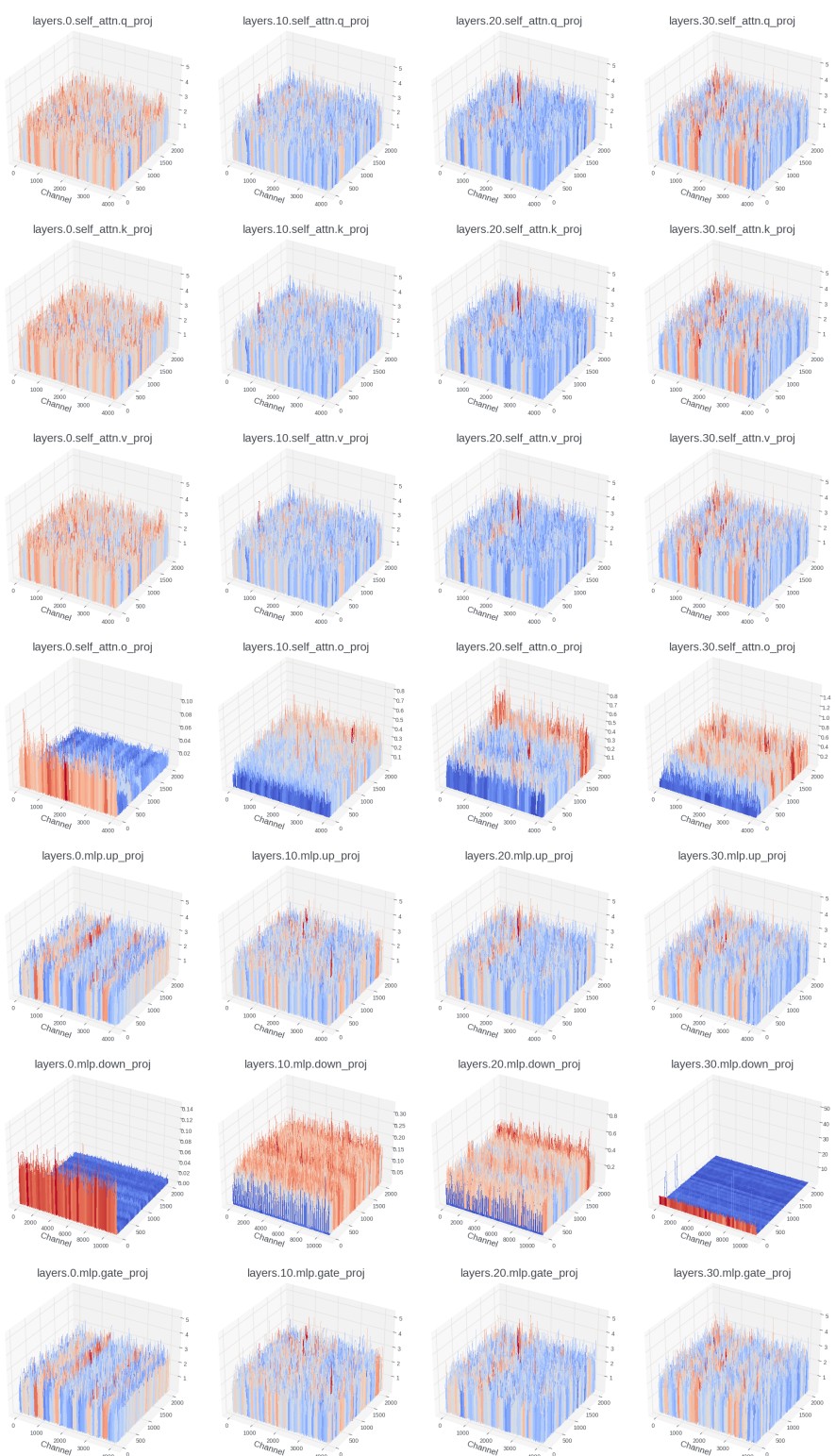

Figure 9: Magnitude of the input activations of a linear layer in $\{1^{st}, 11^{th}, 21^{st}, \text{and } 31^{st}\}$ blocks in LLaMA-2 7B model **after rotation**.

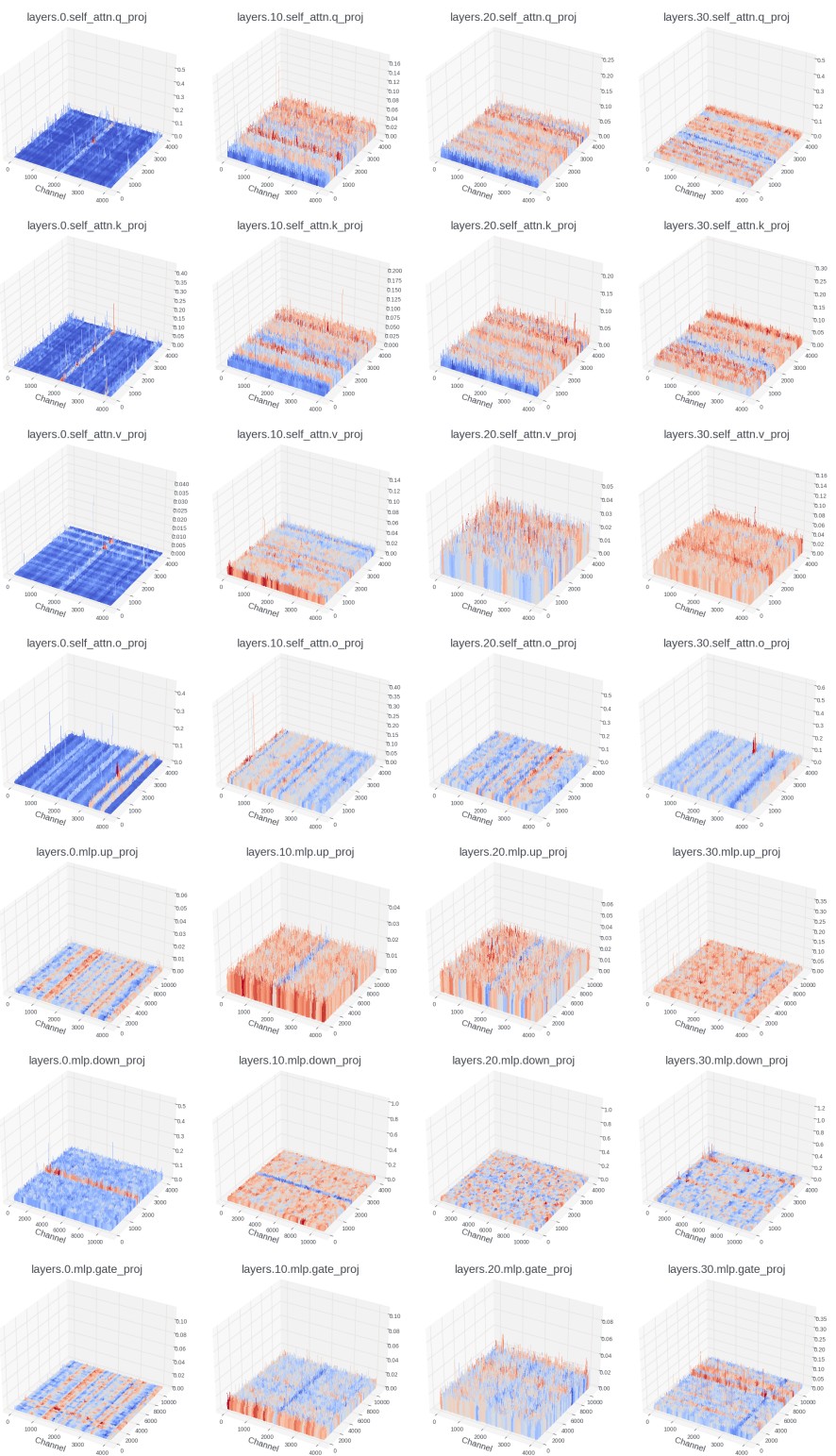

Figure 10: Magnitude of the weights of a linear layer in $\{1^{st}, 11^{th}, 21^{st}, \text{and } 31^{st}\}$ blocks in LLaMA-2 7B **before rotation**.

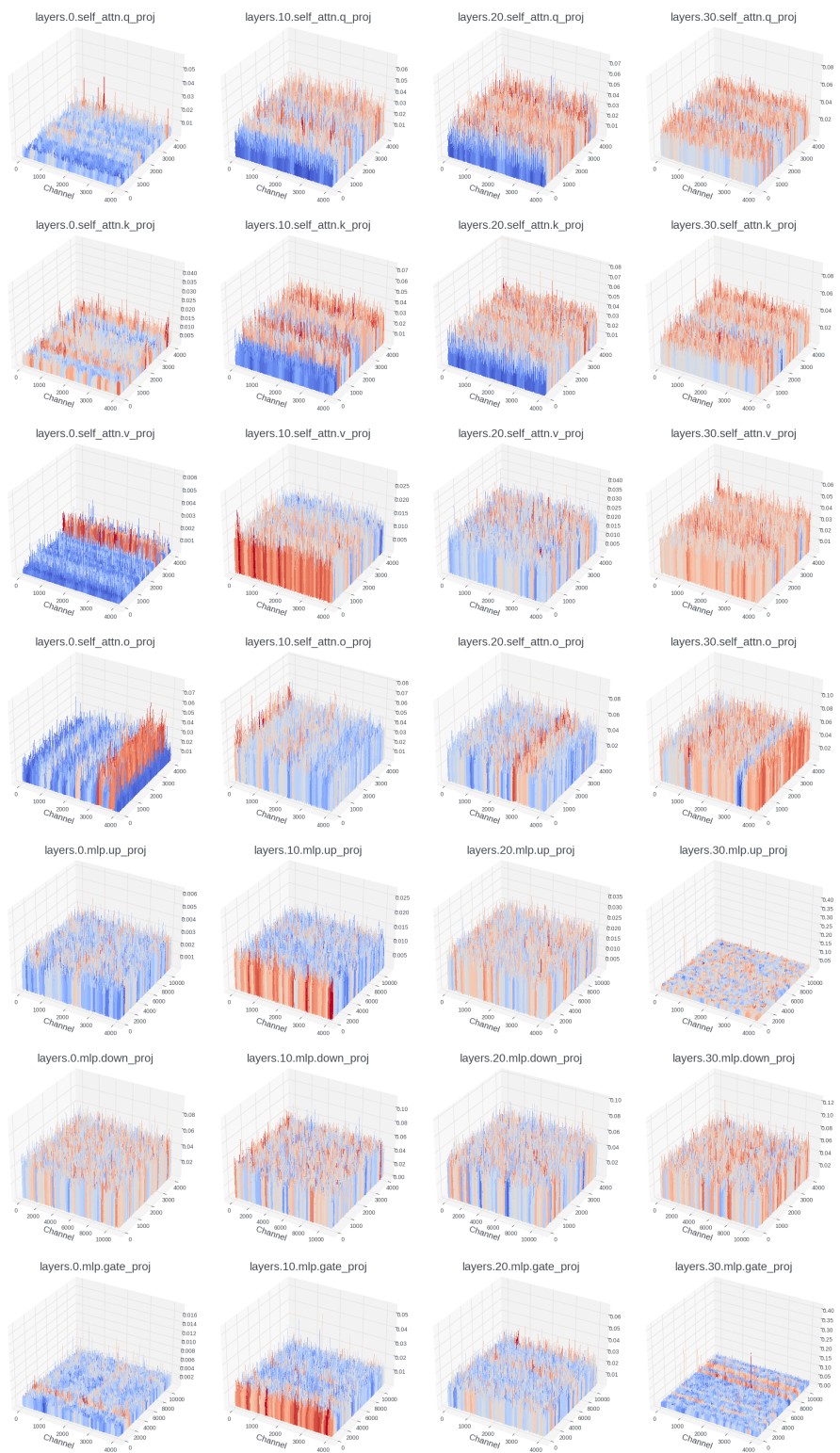

Figure 11: Magnitude of the weights of a linear layer in $\{1^{st}, 11^{th}, 21^{st}, \text{and } 31^{st}\}$ blocks in LLaMA-2 7B **after rotation**.

