# OpenReview forum: "SpinQuant: LLM Quantization with Learned Rotations"
_ICLR.cc/2025/Conference — ICLR 2025 Poster_

### Official Review · Reviewer_brBn · 2024-10-28

**Soundness:** 3
**Presentation:** 3
**Contribution:** 3
**Rating:** 6
**Confidence:** 4

**Summary:**

This paper introduces SpinQuant, a novel approach to mitigate outliers during Large Language Model (LLM) quantization by leveraging learned rotation matrices. The paper presents two rotation strategies: 1) an "easy mode" for W4A8KV8 quantization where rotation matrices can be absorbed without impacting inference, and 2) a "hard mode" incorporating two additional online rotation matrices for more aggressive quantization scenarios (e.g., W4A4KV4). Through evaluation across various LLMs and reasoning tasks, SpinQuant demonstrates that its learned rotation matrices outperform random rotation approaches.

**Strengths:**

- Proposed a learned version of rotation matrices to address variance issues inherent in random rotation approaches, with demonstrated flexibility across different quantization scenarios.
- The paper is well-structured, presenting clear methodology and comprehensive diagrams that effectively illustrate the proposed approach.

**Weaknesses:**

- Limited evaluation scope on complex tasks benchmarks: The evaluation would be more compelling with results from more challenging benchmark tasks, such as [MMLU](https://arxiv.org/abs/2009.03300) and [MMLU-Pro](https://arxiv.org/abs/2406.01574).
- Incomplete theoretical analysis, while Section 4.4 demonstrates Hadamard rotation's superiority over random rotation, it lacks a thorough comparative analysis between learned and random rotation matrices. The significant 10+ point improvement observed on Mistral-7B using hard mode could serve as a good case study to demonstrate the benefits of learned rotation matrices.
- Limited performance benchmarking: The end-to-end speed evaluation is confined to MacBook M1 hardware, with only basic FP8 quantization latency testing on GPU rather than comprehensive benchmarking across the paper's proposed quantization scenarios. Given recent research highlighting potential performance impacts of online rotation transformations (e.g., [QServe]( https://arxiv.org/abs/2405.04532), Part V Section C), more extensive benchmarking on modern GPU architectures would strengthen the paper's practical implications. Key speedup metrics could include time to first token (TTFT) and tokens per second (TPS).

**Questions:**

- Regarding Formula (1), is the clip operation missing?
- For Figure 1, there are two R3 after the `RoPE`, should one of them be the inverse of R3?
- Could you clarify which rotation type is being demonstrated in Figures 9 and 10?

---

> ### Author Response · Authors · 2024-11-19
>
> Thank you for your insightful feedback. We have incorporated your suggestions into the revision to make paper more compelling. Below is a detailed response to each of your questions.
>
> **Weaknesses:**
>
> **W1:** Limited evaluation scope on complex tasks benchmarks: The evaluation would be more compelling with results from more challenging benchmark tasks, such as MMLU and MMLU-Pro.
>
> **A1:** Thanks for the suggestion. We added comparisons on MMLU, using the LLaMA 3.2 1B and 3B models. Our findings indicate that SpinQuant W4A8 quantization significantly enhances performance on 5-shot MMLU evaluation tasks and substantially narrows the performance gap relative to the BF16 baseline:
>
> |  | llama3.2 1B |  |  | llama3.2 3B |  |  |
> | - | - | - | - | - | - | - |
> |  | BF16 | Vanilla RTN | SpinQuant | BF16 | Vanilla RTN | SpinQuant |
> | MMLU (5-shot) | 49.3 | 43.4 | 47.3 | 63.4 | 60.5 | 62.0 |
>
> We added these results on challenging benchmarks to the revision.
>
> **W2:** Incomplete theoretical analysis, while Section 4.4 demonstrates Hadamard rotation's superiority over random rotation, it lacks a thorough comparative analysis between learned and random rotation matrices. The significant 10+ point improvement observed on Mistral-7B using hard mode could serve as a good case study to demonstrate the benefits of learned rotation matrices.
>
> **A2:** Thank you for recommending an ablation study on the Mistral 7B model to evaluate the impact of learned rotation matrices. We are pleased to let you know that this analysis is already included in the paper. Table 2 presents a comprehensive ablation study that highlights the effectiveness of learning rotation matrices. Specifically, for the Mistral-7B model, learning the rotation achieves a more than 15 percentage points in zero-shot accuracy compared to using random Hadamard rotations.
>
>
> **W3:** Limited performance benchmarking: The end-to-end speed evaluation is confined to MacBook M1 hardware, with only basic FP8 quantization latency testing on GPU rather than comprehensive benchmarking across the paper's proposed quantization scenarios. Given recent research highlighting potential performance impacts of online rotation transformations (e.g., QServe, Part V Section C), more extensive benchmarking on modern GPU architectures would strengthen the paper's practical implications. Key speedup metrics could include time to first token (TTFT) and tokens per second (TPS)
>
> **A3:** Thank you for your insightful question. If we understand correctly, the primary reason in Qserve paper that the rotation-based method did not achieve optimal speed is not because of the present of rotation matrix but the assumption made by Qserve – W4A4 rotation quantization must be paired with group-wise quantization to attain high accuracy. Unfortunately, this approach lacks **Tensor core** support and relies solely on **CUDA core** support, which is significantly more expensive—approximately 50 times more costly than using Tensor cores.
>
> In contrast, SpinQuant employs token-wise quantization for all its quantization results. Thus SpinQuant can benefit from **Tensor core** support, allowing for more efficient processing and potentially better accuracy outcomes.
>
> In light of the available **Tensor cores** in NVIDIA's Hopper (H100) architecture, in the rebuttal we provide the **whole network end-to-end** speed test result of W-fp8-A-fp8 quantization on H100 GPU, both with and without Hadamard transformations. Specifically, we utilize FP8 GEMM from the FBGEMM repo [2], which incorporates dequantization via epilogue fusion. We also leverage the Tensor Core-based Hadamard transform kernel [3] to minimize the overhead of the online Hadamard transform. Looking ahead, we plan to conduct a comprehensive speed and accuracy benchmark on W-fp4-A-fp4 quantization upon the arrival of the Blackwell (B100), which will support fp4 Tensor Core operations [4]. At present, fp8 quantization remains the only format capable of delivering reasonable GPU performance on Hopper architecture (H100), as other formats necessitate prolonged kernel development cycles or fail to yield meaningful results. The end-to-end speed test results of LLaMA-3 70B are detailed below:
>
> ||SpinQuant without hadamard||SpinQuant with hadamard||
> |-|-|-|-|-|
> ||TTFT (ms)|TPS (ms)|TTFT (ms)|TPS (ms)|
> |BS=1 T=4096|153.58|9.85|158.25|10.15|
> |BS=8 T=4096|1205.47|10.6|1243.48|10.94|
> |BS=32 T=4096|5008.25|13.83|5147.59|14.24 |
>
> When implemented meticulously, SpinQuant with Hadamard rotation sees marginal difference in the latency compared to without Hadamard rotation. We included these results in the revision.
> [1] www.exxactcorp.com/blog/hpc/comparing-nvidia-tensor-core-gpus
> [2] https://github.com/pytorch/FBGEMM/blob/main/fbgemm_gpu/experimental/gemm/triton_gemm/fp8_gemm.py
> [3] https://github.com/pytorch-labs/applied-ai/blob/main/kernels/cuda/inference/hadamard_transform/hadamard_transform.cpp
> [4] https://resources.nvidia.com/en-us-blackwell-architecture

---

> ### Author Response · Authors · 2024-11-19
>
> **Questions:**
>
> **Q1:** Regarding Formula (1), is the clip operation missing?
>
> **A1:**  Since we employ min-max quantization, the values of X are scaled to the range $(-2^{N-1} -1, 2^{N-1} -1)$, where N is number of bits. Therefore, clipping is not necessary in this context.
>
> **Q2:** For Figure 1, there are two R3 after the RoPE, should one of them be the inverse of R3?
>
> **A2:** We appreciate your careful reading of our paper. The use of R3 is correct because the operation $(W_q @ R_3) @ (W_k @ R_3)^T$ $= W_q @ R_3 @ R_3^T @ W_k^T$ $= W_q @ W_k$ holds true. Thus, it should indeed be R3 in the figure, rather than $R_3^{-1}$ or $R_3^T$.
>
> **Q3:** Could you clarify which rotation type is being demonstrated in Figures 9 and 10?
>
> **A3:** Figures 9 and 10 demonstrate the SpinQuant optimized rotations. We clarified it in the paper.

---

> > ### Comment · Reviewer_brBn · 2024-12-02
> >
> > Thank you for addressing my questions and providing additional clarification.

---

> ### Author Response · Authors · 2024-11-25
>
> Dear Reviewer brBn,
>
> Thanks for your positive initial review. If our response effectively resolves your issue, we'd be grateful if you consider further improving your score.
>
> Thank you!
> Authors

---

### Official Review · Reviewer_tcZR · 2024-11-04

**Soundness:** 3
**Presentation:** 3
**Contribution:** 3
**Rating:** 6
**Confidence:** 2

**Summary:**

This paper introduces SpinQuant, a PTQ method that applies optimized rotations to reduce quantization errors from outliers, improving LLM efficiency without compromising accuracy. It achieves near full-precision performance, narrowing the accuracy gap to 2.9 points on zero-shot tasks in LLaMA-2 7B, outperforming previous works.

**Strengths:**

- paper is well-written
- specifically, the illustrative example in 4.4 is very helpful. Yet, it would be sensible to move this one more to the front of the paper
- experiments are extensive across models, settings and methods

**Weaknesses:**

- The current rotation is not optimal, as the authors state. It would be a very valuable contribution to investigate optimal rotations further.
- a more detailed discussion on model size vs rotation-based quantization is needed. Larger models are impacted less by quantization compared to smaller models, why?
- currently, no effort to publish code or quantized models given, limiting reproducibility and usability for the community.
- the tradeoff discussion between easy and hard is too shallow. Please elaborate on which cases are useful to use one or the other.

**Questions:**

- Could you provide a comparison of the computation needed for the quantization? Going beyond comparing accuracies on benchmarks alone would be helpful.
- the number of rotation matrices is determined empirically. Can the authors investigate this further and explain in more detail how this can be validated better?

---

> ### Author Response · Authors · 2024-11-19
>
> We appreciate your insightful feedback. We have integrated your suggestions into the revision, which further enhances the paper's quality. Below is a detailed response to each of your questions.
>
> **Weaknesses:**
>
> **W1:** The current rotation is not optimal, as the authors state. It would be a very valuable contribution to investigate optimal rotations further.
>
> **A1:** Thank you for your comments. In our paper, we point out that using random rotations is not optimal and that learned rotations can yield significantly better results. Although the learned rotation does not have a theoretical proof of optimality, it can greatly reduce the variance introduced by random rotation and is a valuable solution with high accuracy.
>
> **W2:** A more detailed discussion on model size vs rotation-based quantization is needed. Larger models are impacted less by quantization compared to smaller models, why?
>
> **A2:** Thank you for the insightful question. It is a widely observed trend that larger models are less impacted by quantization due to their inherent redundancy. This observation is supported by works such as LLM-QAT [1] and GPTQ [2]. Although some early studies [3] in the LLM era found that larger models preserve more outliers, making them initially harder to quantize, these outlier issues have been addressed in various ways. Consequently, larger models with more neurons experience less accuracy drop compared to smaller models. This trend of larger models being less affected by quantization is also observed in ViT compression works [4][5].
> [1] Llm-qat: Data-free quantization aware training for large language models.
> [2] GPTQ: Accurate Post-Training Quantization for Generative Pre-trained Transformers.
> [3] SmoothQuant: Accurate and Efficient Post-Training Quantization for Large Language Models.
> [4] Oscillation-free Quantization for Low-bit Vision Transformers.
> [5] Q-ViT: Accurate and Fully Quantized Low-bit Vision Transformer.
>
>
> **W2:** currently, no effort to publish code or quantized models given, limiting reproducibility and usability for the community.
>
> **A3:** We have already released the code and models. These resources are publicly available, but we cannot include the links in the paper submission or rebuttal to maintain anonymity during the review process.
>
> **W4:** the tradeoff discussion between easy and hard is too shallow. Please elaborate on which cases are useful to use one or the other.
>
> **A4:** Thank you for highlighting this aspect. SpinQuant-easy is designed to facilitate low-bit weight quantization and 8-bit activation quantization without the need for online Hadamard transformations. This approach allows for seamless integration by simply substituting the original model weights with the rotated model weights, eliminating the necessity for modifications in the forward pass or additional kernel support. In practical deployment scenarios, SpinQuant-easy offers a cost-free quantization accuracy boost, making it an ideal choice for projects with stringent design timelines. Experimenting with SpinQuant-easy does not compromise inference speed and is a worthwhile consideration for improving quantization quality.
>
> On the other hand, SpinQuant-hard leverages R3 and R4 online rotations, as depicted in Figure 1. The R4 rotation is particularly effective in mitigating activation outliers within the down projection layer of the MLP, and R3 effectively addresses outliers in the KV-cache, thereby substantially improving performance in low-bit activation quantization and KV-cache quantization scenarios, respectively. SpinQuant-hard is the optimal solution for applications where precision is critical and is compatible with devices that support Hadamard kernels. It is also well-suited for projects with longer development timelines that can incorporate kernel integration. For activation quantization below 8-bit, SpinQuant-hard is the recommended choice to achieve unparalleled accuracy. We will extend these discussions in the revision.

---

> ### Author Response · Authors · 2024-11-19
>
> **Questions:**
>
> **Q1:** Could you provide a comparison of the computation needed for the quantization? Going beyond comparing accuracies on benchmarks alone would be helpful.
>
> **A1:** Certainly. In lines 301-303 of the paper, we have detailed the optimization time for each model. In the table below, we show the optimization time comparison between GPTQ and SpinQuant. SpinQuant requires a similar scale of optimization time as GPTQ. The additional optimization time required by SpinQuant is worthwhile considering the substantial improvements it offers over GPTQ. We added these discussions in the revision
>
>  | | llama3 1B | llama3 3B | llama3 8B | llama2 7B | llama2 13B | Mistral |
> | - | - | - | - | - | - | - |
>  | SpinQuant | 13 min | 18 min | 30 min | 25 min | 30 min | 16 min |
>  | GPTQ | 8 min | 13 min | 20 min | 18 min | 25 min | 12 min |
>
> **Q2:** the number of rotation matrices is determined empirically. Can the authors investigate this further and explain in more detail how this can be validated better?
>
> **A2:** Thanks for the question. The number of rotation matrices is not determined empirically but is strategically designed to mitigate outliers before every activation quantization. Specifically, R1 addresses activation outliers in the input activations to Q, K, V, and up projection and gate projection fully-connected layers. R2 targets outliers in the input activation to the O projection layer in the self-attention block, and R4 further addresses outliers in input activations to the down projection layer in the MLP block. As shown in Figure 1, these rotations are inserted before every fully-connected layer to ensure that input activations are optimally rotated for easier quantization. Besides R124, R3 rotates the KV cache before KV cache quantization to alleviate the outlier.

---

> > ### Comment · Reviewer_tcZR · 2024-11-20
> >
> > Thanks, can you please provide a time comparison to QuIP#?

---

> > > ### Author Response · Authors · 2024-11-23
> > >
> > > Thanks for the follow-up question. Do you mean optimization time or inference time?

---

> > > > ### Comment · Reviewer_tcZR · 2024-11-26
> > > >
> > > > preferably both

---

> > > > > ### Author Response · Authors · 2024-11-27
> > > > >
> > > > > Thank you for the clarification.
> > > > >
> > > > > **Optimization time**
> > > > >
> > > > > We conducted an optimization time analysis using an example w4 script for the llama-2-7b-hf model (E8P12RVQ4B codebook) with the original codebase available at Cornell-RelaxML/quip-sharp. According to their paper, the QuIP# optimization process comprises three stages:
> > > > >
> > > > > - Stage 1:  Hessian Computation (Section 4)
> > > > > - Stage 2:  Fine-tuning within Each Transformer Block (Section 5)
> > > > > - Stage 3:  Minimizing Error Over the Entire Model (Section 5)
> > > > >
> > > > > We benchmarked the time required for each stage of the llama-2-7B model as follows:
> > > > >
> > > > > - Stage 1: 73 minutes
> > > > > - Stage 2: 51 minutes
> > > > > - Stage 3: Unfortunately, we were unable to execute this experiment due to the codebase being outdated.
> > > > >
> > > > > In comparison, SpinQuant requires only 25 minutes to optimize the llama-2 7B model, making it approximately five times faster than QuIP#—even without including the time for Stage 3.
> > > > >
> > > > > **Inference Time**
> > > > >
> > > > > We currently don’t have the kernel for W4A16 on GPU, so we are putting together the relevant kernels for the speed testing. We will post the inference time comparison in a couple of days!

---

> > > > > > ### Comment · Reviewer_tcZR · 2024-12-02
> > > > > >
> > > > > > Thanks for the responses. I vote for accepting this paper.

---

### Official Review · Reviewer_U4Hi · 2024-11-05

**Soundness:** 2
**Presentation:** 3
**Contribution:** 2
**Rating:** 6
**Confidence:** 4

**Summary:**

The paper presents SpinQuant, a quantization technique specifically designed to reduce the impact of outliers in large language models (LLMs) through learned rotations. SpinQuant introduces learned rotation matrices that can be integrated into pre-trained weights, which aids in preserving accuracy in low-bit quantization scenarios. The proposed method includes two modes: SpinQuant-easy and SpinQuant-hard—the former for more straightforward quantization with mergeable weights, and the latter for complex scenarios needing enhanced outlier suppression with non-mergeable weights. The authors validate SpinQuant across multiple models (e.g., LLaMA-2, LLaMA-3, and Mistral) using a comprehensive set of zero-shot reasoning benchmarks.

**Strengths:**

- By employing learned rotations, SpinQuant offers a new approach to minimizing outliers.
- The SpinQuant-easy and SpinQuant-hard modes offer flexibility, making the approach adaptable to different computational constraints and accuracy requirements through mergeable and non-mergeable weights.
- SpinQuant shows compatibility with methods like GPTQ, allowing it to integrate into established quantization pipelines.

**Weaknesses:**

- The benchmarks and model selections, such as the LLaMA series, seem to focus on favorable cases for SpinQuant. Testing on a broader range of architectures (e.g., Gemma2) would provide a more thorough evaluation. Especially Gemma2 series which exhibit distinct activation characteristics—could provide a more comprehensive assessment of SpinQuant's generalizability and robustness across diverse LLM types.
- Although the authors note the optimization time (up to 3.5 hours for larger models), it remains unclear how this time impacts real-world deployment, especially on low-resource devices.
- From an outlier-aware quantization perspective, as demonstrated by methods like AWQ, SpinQuant should ideally exhibit strong performance even in weight-only quantization scenarios. However, even after examining the appendix, there is no evaluation comparing its performance under weight-only quantization. It would be beneficial to include a comparison of SpinQuant’s performance in weight-only quantization settings to provide a more comprehensive view of its robustness and effectiveness relative to other quantization techniques.
- While the paper presents promising results on zero-shot reasoning tasks, it does not evaluate SpinQuant's performance in fine-tuning or few-shot learning scenarios. Including such evaluations would strengthen the claims about SpinQuant’s overall effectiveness and versatility.

**Questions:**

- Have the authors considered testing alternative calibration datasets beyond WikiText2? It would be interesting to see if different calibration sets impact SpinQuant’s generalization, especially in domain-specific applications (e.g., code). While the paper describes the use of WikiText2 for calibration, further analysis on its impact and comparison with alternative calibration data could strengthen the understanding of its generalization.
- it would be interesting to know if SpinQuant has been tested on models from other domains, such as Vision Transformers (ViTs) or vision-language models, to further assess its versatility across different types of architectures.

---

> ### Author Response · Authors · 2024-11-19
>
> Thank you for your insightful comments. Based on your suggestions, we have made the following updates: (1) extended SpinQuant to more models, including Phi-3, (2) included results for weight-only quantization, (3) evaluated SpinQuant's performance in fine-tuning and few-shot learning scenarios, and (4) tested alternative calibration datasets beyond WikiText2. In all these experiments, SpinQuant consistently outperforms the baseline. We have added these results to the revision to make SpinQuant more comprehensive, and we hope you will consider increasing the score in light of SpinQuant's strong results. Please let us know if you have any follow-up questions.
>
>
> **Weaknesses:**
>
> **W1:** The benchmarks and model selections, such as the LLaMA series, seem to focus on favorable cases for SpinQuant. Testing on a broader range of architectures (e.g., Gemma2) would provide a more thorough evaluation. Especially Gemma2 series which exhibit distinct activation characteristics—could provide a more comprehensive assessment of SpinQuant's generalizability and robustness across diverse LLM types.
>
> **A1:** Thank you for your feedback. First, in addition to the LLaMA series, we have also evaluated SpinQuant on the Mistral models, as shown in Table 1 of the original paper. SpinQuant demonstrates consistently significant improvements on Mistral models as well, which are quite different from LLaMA models.
>
> Regarding the implementation of SpinQuant in Gemma2 models, we encountered a unique challenge. Gemma2 employs both pre-norm and post-norm structures  (x $\rightarrow$ pre-norm $\rightarrow$ FFN//MHSA $\rightarrow$ pre-norm $\rightarrow$ + x). The post-layer normalization cannot be absorbed into the corresponding weights due to the presence of  $\frac{x}{||x||}$ between the last fully connected layer and the channel-wise scale in the norm layer. This limitation prevents the network from fully absorbing rotation matrices into the weight matrices, allowing only the application of R1 as online rotation matrices. While feasible, this approach results in an architecture that deviates from the original paper and should be considered an extension of the SpinQuant method. Therefore, we did not include the results here.
>
> Most existing large language models (LLMs) utilize a pre-norm architecture, making them compatible with SpinQuant. These LLM models include but are not limited to the Phi series, OPT, Pythia, Qwen, Yi, etc. Consequently, we conducted further experiments on the Phi-3 model, where SpinQuant also demonstrated consistent improvements over the baseline.
>
> | Phi-3 FP | Phi-3 W4A4 RTN | Phi-3 W4A4 GPTQ | Phi-3 W4A4 SpinQuant |
> | - | - | - |  - |
> | 6.32 | 1627.38 | 340.02 | 7.89 |
>
> These results, along with the Mistral findings presented in the original paper, offer a more comprehensive assessment of SpinQuant's generalizability and robustness across diverse LLM types. We will include the Phi-3 results in the revision to provide a more complete picture.
>
> **W2:** Although the authors note the optimization time (up to 3.5 hours for larger models), it remains unclear how this time impacts real-world deployment, especially on low-resource devices.
>
> **A2:**  In actual efficient inference deployment scenarios, the common approach is to use the GPU for offline model compression, and then deploy the compressed model to the target device. This is our target scenario. Our paper shows that obtaining compressed models offline is inexpensive and that quantization can significantly improve inference speed. While on-device optimization/training is an interesting topic, our focus is on optimizing on-device inference. We will clarify this point in the revised paper.

---

> ### Author Response · Authors · 2024-11-19
>
> **W3:**  From an outlier-aware quantization perspective, as demonstrated by methods like AWQ, SpinQuant should ideally exhibit strong performance even in weight-only quantization scenarios. However, even after examining the appendix, there is no evaluation comparing its performance under weight-only quantization. It would be beneficial to include a comparison of SpinQuant’s performance in weight-only quantization settings to provide a more comprehensive view of its robustness and effectiveness relative to other quantization techniques.
>
> **A3:** Thank you for highlighting this aspect. In response, we have included a comparison of SpinQuant’s performance under weight-only quantization in the table below:
>
> | Method | LLaMA-2 7B | | LLaMA-2 13B | | LLaMA-2 | |
> |-|-|-|-|-|-|-|
> | | Zero-shot Avg. | Wiki ppl | Zero-shot Avg. | Wiki ppl | Zero-shot Avg. | Wiki ppl |
> | RTN | 63.6 | 7.2 | 57.9 | 6.4 | 69.2 | 4.6 |
> | SmoothQuant | 59.1 | 7.5 | 63.3 | 6.1 | 70.2 | 4.1 |
> | GPTQ | 64.5 | 11.3 | 64.7 | 5.6 | 71.9 | 3.9 |
> | AWQ | -- | 6.2 | -- | 5.1 | -- | -- |
> | **SpinQuant**| **65.9** | **5.6** | **68.5** | **5.0** | **72.6** | **3.5** |
>
> The weight-only results shows SpinQuant consistently achieves higher accuracy than AWQ and other previous works. We included these results in revision to provide a more comprehensive view.
>
>
> **W4:**  While the paper presents promising results on zero-shot reasoning tasks, it does not evaluate SpinQuant's performance in fine-tuning or few-shot learning scenarios. Including such evaluations would strengthen the claims about SpinQuant’s overall effectiveness and versatility.
>
> **A4:** Thanks for the suggestion. We added experiments applying SpinQuant to **instruction-finetuned** LLaMA 3.2 1B and 3B models. We present the results for few-shot learning scenarios performance below. SpinQuant W4A8 quantized models demonstrate significant improvements in 5-shot accuracy on the MMLU benchmark and 1-shot rouge score on the TLDR9 summarization benchmark. It closed the gap to the BF16 baseline significantly. We added these findings to the paper to further strengthen SpinQuant’s overall effectiveness and versatility.
>
> |  | llama3.2 1B |  |  | llama3.2 3B |  |  |
> | - | - | - | - | - | - | - |
> |  | BF16 | Vanilla RTN | SpinQuant | BF16 | Vanilla RTN | SpinQuant |
> | MMLU (5-shot) | 49.3 | 43.4 | 47.3 | 63.4 | 60.5 | 62.0 |
> | TLDR9+ (test, 1-shot rougeL) | 16.8 | 14.9 | 16.7 | 19.0 | 19.1 | 19.2 |
>
> **Questions:**
>
> **Q1:** Have the authors considered testing alternative calibration datasets beyond WikiText2? It would be interesting to see if different calibration sets impact SpinQuant’s generalization, especially in domain-specific applications (e.g., code). While the paper describes the use of WikiText2 for calibration, further analysis on its impact and comparison with alternative calibration data could strengthen the understanding of its generalization.
>
> **A1:** Thanks for the question. We had that experiment in Table 13 in the appendix. We conducted experiments on the LLaMA-2 7B model using the C4 dataset for calibration. Our findings indicate that SpinQuant is robust to the choice of calibration data, yielding similar results with C4 as with the Wiki dataset. The results are presented below:
>
> | Data | #Bits | ARC-e | ARC-c | BoolQ | PIQA | SIQA | HellaS. | OBQA | WinoG. | Avg. | Wiki |
> |-|-|-|-|-|-|-|-|-|-|-|-|
> | Wiki | W4-A4-KV16 | 72.1 | 47.5 | 74.4 | 77.0 | 47.3 | 73.2 | 54.4 | 66.9 | 64.1 | 5.9 |
> | | W4-A4-KV16 | 72.6 | 47.5 | 73.9 | 77.0 | 47.2 | 73.0 | 54.1 | 66.9 | 64.0 | 5.9 |
> | C4 | W4-A4-KV16 | 72.5 | 47.3 | 74.8 | 77.6 | 47.7 | 73.2 | 55.4 | 66.2 | 64.3 | 5.9 |
> | | W4-A4-KV4 | 72.5 | 47.9 | 74.0 | 78.4 | 46.7 | 73.1 | 55.5 | 66.4 | 64.3 | 6.0 |
>
>
> **Q2:** it would be interesting to know if SpinQuant has been tested on models from other domains, such as Vision Transformers (ViTs) or vision-language models, to further assess its versatility across different types of architectures.
>
> **A2:**  We appreciate your suggestion to explore SpinQuant's applicability to models from other domains, such as Vision Transformers (ViTs) or vision-language models. This is indeed an interesting direction for future work. While our current study focuses on the language modality, we will include a discussion of potential future work in the paper to explore SpinQuant's versatility across different modalities.

---

> ### Author Response · Authors · 2024-11-25
>
> Dear Reviewer U4Hi,
>
> For your queries, we carefully prepared the answers. We look forward to you considering raising your score if our response effectively addressed your question.
>
> Thank you!
> Authors

---

> ### Comment · Reviewer_U4Hi · 2024-11-26
> **Response to the Authors feedback.**
>
> Thank you for your detailed and thoughtful responses. The updates and additional experiments significantly address several of the points raised in the initial review. Below are my reflections on the revisions and responses:
>
> ----
>
> **1. Presentation of Results:**
> I noticed that some experimental tables include empty cells for certain baselines (e.g., Table1 OmniQuant and QuIP, and AWQ). While I understand that this might be due to the unavailability of results from the original papers, presenting complete comparisons is crucial for clarity and thoroughness.
>
> **2. Broader Architecture Testing (W1):**
> The inclusion of Phi-3 results strengthens the generalizability claims of SpinQuant across diverse LLM architectures. Regarding the challenges with Gemma2 models due to the pre-norm and post-norm architecture, I understand that this limitation makes it more appropriate to consider such adaptations as future work. Including this clarification in the revised paper will provide readers with a clear understanding of SpinQuant’s current scope and highlight potential avenues for future research.
>
> **3. Weight-Only Quantization (W3):**
> The newly included weight-only quantization results are a valuable addition and demonstrate SpinQuant's robustness in this scenario. While the weight-only quantization results are valuable, it would enhance the evaluation to include other metrics or tasks beyond zero-shot evaluations.  This would provide a more complete picture of SpinQuant's capabilities in weight-only setups and broaden the scope of its demonstrated effectiveness.
>
> **4. MMLU and TLDR9+ (W4):**
> The additional experiments on the MMLU and TLDR9 benchmarks, provide fair evidence of SpinQuant’s versatility. These results significantly enhance the paper’s claims about its applicability beyond zero-shot reasoning tasks. Including these findings in the revision effectively addresses the concern about broader applicability.
>
> ----
>
> While it is unfortunate that some results in the tables are incomplete or that certain comparisons were not included, I acknowledge the novelty of the proposed method (even considering concurrent work) and the substantial efforts made by the authors to address these limitations. I trust that these gaps will be fully addressed in the revised version to ensure a more complete and balanced evaluation.
>
> Given the overall quality of the revisions and the promising contributions of the method, I have decided to raise my score, believing that the authors will make the necessary adjustments to further enhance the clarity and comprehensiveness of the paper.

---

> > ### Author Response · Authors · 2024-11-27
> >
> > Thank you for your detailed response and valuable suggestions.
> >
> > Yes, we will incorporate a comprehensive comparison of these models, scenarios, and benchmarks in our revision. We will also clarify the applicable range of SpinQuant to provide a complete picture of its broader applicability, such that future research can build on top of it and further extend the capability boundaries of rotation-based quantization methods.

---

### Official Review · Reviewer_Jf43 · 2024-11-06

**Soundness:** 4
**Presentation:** 3
**Contribution:** 3
**Rating:** 6
**Confidence:** 3

**Summary:**

This work investigates post-training quantization (PTQ) for LLMs, applied to weights, activations, and KV cache. It addresses the large quantization errors issue from outliers by introducing the learned rotation matrices into the network architecture. Through experiment validations, the proposed framework shows effectiveness in extreme W4A4KV4 quantization settings on representative networks.

**Strengths:**

+ The paper pursues the rotation method for post training LLM quantization, which demonstrates convincing performance.

+ The authors conducted thorough investigations about random rotations to propose quantization-oriented rotation learning.

+ The proposed framework develops strategies for the challenging activation and KV cache quantization, besides weight quantization.

**Weaknesses:**

- While the rotation matrix method is effective in mitigating outliers in LLM quantization, it is better if more rigorous explanations/proofs are provided for the rationale behind, besides empirical results.

- Introducing rotation matrices into the inference pipeline may lead to overhead in computing. Could the authors provide analysis the overhead vs benefit from quantization?

**Questions:**

Although the rotation method is effective, the readers may be interested in the theoretical foundations of this methodology.

Please provide analysis on the overhead introduced by the rotation matrices into the inference pipeline vs the benefits.

The reviewer would like to know the training complexity of the proposed method comparing with other quantization approaches.

---

> ### Author Response · Authors · 2024-11-19
>
> Thank you for your valuable feedback. We have incorporated your suggestions into the revision, which helps further improve the quality of the paper. Below are point-by-point responses to your questions.
>
> **Weaknesses:**
>
> **W1:** While the rotation matrix method is effective in mitigating outliers in LLM quantization, it is better if more rigorous explanations/proofs are provided for the rationale behind, besides empirical results.
>
> **A1:** Thanks for the suggestion. The core idea is that outliers can significantly impact quantization performance. For instance, adding an outlier at twice the current distribution maximum can effectively reduce the bit-width by one. The Hadamard transformation statistically bounds outliers with a certain probability, specifically, $max(|W_{i,j}|) \leq \mu ||W||_F / \sqrt{mn}$ with a probability of $1 - \delta$, where $\mu = 2 \log (4mn/\delta)$, as described in QuIP# [1]. While this theoretical bound is not particularly tight, outliers may still occur, our method addresses this by optimizing the rotation matrix to further mitigate these randomly occurring outliers.
> [1] QuIP#: Even Better LLM Quantization with Hadamard Incoherence and Lattice Codebooks
>
> **W2:** Introducing rotation matrices into the inference pipeline may lead to overhead in computing. Could the authors provide analysis of the overhead vs benefit from quantization?
>
> **A2:** Thank you for raising this point. (1) R1/R2 rotation can be incorporated into the weight without any overhead, i.e., SpinQuant-easy can improve the quantization without introducing any additional inference cost. (2) For R3/R4, we did an end-to-end speed test of LLaMA-3 8B in Table 6, the inclusion of the online Hadamard matrix introduces approximately 8% overhead compared to a network without it. This acceptable overhead results in a substantial accuracy improvement of over 10 points compared to previous methods, especially on low-bit activation quantization scenarios, indicating that the benefits of online rotation justify the computational efforts.
>
>
> **Questions:**
>
> **Q1:** The reviewer would like to know the training complexity of the proposed method comparing with other quantization approaches.
>
> **A1:** Certainly. In lines 301-303 of the paper, we have detailed the optimization time for each model. In the table below, we show the optimization time comparison between GPTQ and SpinQuant. SpinQuant requires a similar scale of optimization time as GPTQ. The additional optimization time required by SpinQuant is worthwhile considering the substantial improvements it offers over GPTQ. We added these results and discussions in the revision.
>  | | llama3 1B | llama3 3B | llama3 8B | llama2 7B | llama2 13B | Mistral |
> | - | - | - | - | - | - | - |
>  | SpinQuant | 13 min | 18 min | 30 min | 25 min | 30 min | 16 min |
>  | GPTQ | 8 min | 13 min | 20 min | 18 min | 25 min | 12 min |

---

> ### Author Response · Authors · 2024-11-25
>
> Dear Reviewer Jf43,
>
> Thank you for your positive initial review. If our response effectively resolves your issue, we'd be grateful if you further improve your score.
>
> Thank you!
> Authors

---

### Official Review · Reviewer_6v9t · 2024-11-12

**Soundness:** 2
**Presentation:** 2
**Contribution:** 2
**Rating:** 5
**Confidence:** 4

**Summary:**

The paper presents a learned rotation based method, namely $\texttt{SpinQuant}$ to mitigate outliers in weight and activation distributions, boosting the performance of quantized LLMs. In specific, the authors  proposes optimizing rotation matrices within Stiefel manifold, directly minimizing the final loss of rotated quantized network. The authors further propose two variants, $\texttt{SpinQuant}$_easy and $\texttt{SpinQuant}$_hard without and with change of

**Strengths:**

The idea of learned rotation in  the context of LLM activation outlier reduction seems interesting.

The authors have done thorough analysis to justify the need of learned rotation

$\texttt{SpinQuant}$ can outperform baseline GPTQ by significant margin!

**Weaknesses:**

1. The performance of $\texttt{SpinQuant}$ is similar or worse than QuIP in most of the cases. This further raises concerns on the learned rotation method as QuIP does not rely on any activation rotation method.

2. The calibration discussion (thorough) is missing. As the method needs to learn R1,R2, R3 and R4, such calibration and/or fine-tuning overhead is a concern.

3. How the learning gets affected for activation rotation, when there is extremely low calibration data?

4. The can be other alternatives to learn the R1-4, why did the author chose Cayley SGD method (Li et al., 2020).

5. Why table 6 shows results with $\texttt{SpinQuant}$_hard even for W4A8?

6. What speedup benefit we may have with $\texttt{SpinQuant}$_hard and $\texttt{SpinQuant}$_easy with W4A4?

7. The contribution is primarily centered around rotation to get rid of activation outliers and smoothen out while for the quantization part the authors heavily rely on GPTQ (or RTN). The baseline GPTQ may often be considered a calibration heavy that require Hessian information. I have concerns on the calibration overhead of the method as well as the novelty.

**Questions:**

1. It would be interesting to see the performance of such dynamic quantization ($\texttt{SpinQuant}$) with static quantization alternative [1].

2. Is the R3, R4 needed in case someone uses block reconstruction of [2]?

3. Also, how does $\texttt{SpinQuant}$ compare w.r.t these alternative methods?

4. Why the choice of ``optimizing these rotation matrices on Stiefel manifold", is not fully clear.

[1] PREFIXQUANT: STATIC QUANTIZATION BEATS DYNAMIC THROUGH PREFIXED OUTLIERS IN LLMS, arxiv 2024

[2] TESSERAQ: ULTRA LOW-BIT LLM POST-TRAINING QUANTIZATION WITH BLOCK RECONSTRUCTION, arxiv 2024.

---

> ### Author Response · Authors · 2024-11-19
>
> Thank you, Reviewer 6v9t, for recognizing the novelty of our paper in utilizing learned rotation for LLM activation outlier reduction and for noting its significant performance improvements (>10 points) over the baseline GPTQ.
>
> Your concerns are mainly on (1) the comparison to previous works (QuIP#, GPTQ): in response we show clear benefits compared to them (2) more ablations on calibration data / time and inference time: we added these details (3) the optimization method in Stefiel manifold: we explained more about the concept of orthonormal matrices and Stefiel manifold. Hope our detailed response below can address your concerns and make you favorably raise your scores.
>
> **Weakness**
>
> **W1:** The performance of SpinQuant is similar or worse than QuIP in most of the cases. This further raises concerns on the learned rotation method as QuIP does not rely on any activation rotation method.
>
> **A1:** First, we want to clarify that we have demonstrated significant improvements in the W4A4 scenarios compared to previous works, which was also noted by the Reviewer 6v9t. We want to point out that the performance of SpinQuant **is not inferior to** QuIP\#; in fact, it is **comparable** with QuIP# in W4 case: QuIP\# achieves 5.6/5.0/3.4 wiki-2 perplexity, while SpinQuant achieves 5.7/5.0/3.5 wiki-2 perplexity on LLaMA-2 7B/13B/70B models, respectively.
>
> QuIP\# is specifically designed for weight quantization. It uses: $U^T (quantized(\tilde{W} )(V x))$ $≈U^T (\tilde{W} (V x)) = Wx $ where $\tilde{W} = UWV^T$, $U$ and $V$ are Hadamard matrices. This means QuIP# need to compute two online Hadamard rotations per weight matrix, which generates **significant overhead**. SpinQuant does not have this overhead. We do rotation at network level, as shown in Figure 1, we rotate both weight and activation tensors, and can absorb all R1 R2 matrices into the weight tensor. Our method also tackles activation quantization. Since activations often exhibit more severe outliers than the weights, these are often more problematic. Moreover, QuIP\# uses complicated vector quantization, while SpinQuant uses simple integer quantization.
>
> SpinQuant shows its effectiveness in achieving high accuracy (more than 10 points improvement than baseline) in W4A4 cases, and achieving on-par accuracy but significantly higher efficiency (less online rotation, simple INT quantization) than QuIP\# in W4 cases.
>
> **W2:** The calibration discussion (thorough) is missing. As the method needs to learn R1,R2, R3 and R4, such calibration and/or fine-tuning overhead is a concern.
>
> **A2:** Thanks for the question. We presented a comprehensive ablation study on the number of training samples and iterations affecting accuracy, as detailed in Table 11 of the appendix. SpinQuant is robust to the calibration choices.
>
> In lines 301-303 of the paper, we have detailed the optimization time for each model. In the table below, we show the optimization time comparison between GPTQ and SpinQuant. **SpinQuant requires a similar scale of optimization time as GPTQ**. The additional optimization time required by SpinQuant is worthwhile considering the substantial improvements it offers over GPTQ.
>
>  | | llama3 1B | llama3 3B | llama3 8B | llama2 7B | llama2 13B | Mistral |
> | - | - | - | - | - | - | - |
>  | SpinQuant | 13 min | 18 min | 30 min | 25 min | 30 min | 16 min |
>  | GPTQ | 8 min | 13 min | 20 min | 18 min | 25 min | 12 min |
>
>
> **W3:** How the learning gets affected for activation rotation, when there is extremely low calibration data?
>
> **A3:** Thank you for your insightful question. We have addressed this in Table 11, where we present an ablation study on the number of calibration data points used. The results demonstrate that SpinQuant maintains robustness despite variations in the amount of calibration data. **Even with as few as 128 examples, reliable results can still be achieved.** We argue that random rotation matrices are only approximately removing outliers from their respective distributions, and not much adjusting is needed to get rid of the significant outliers more robustly. And as outliers are disproportionally detrimental to network performance, getting rid of the most egregious ones left after the random rotations improves performance significantly.

---

> ### Author Response · Authors · 2024-11-19
>
> **W4:** There can be other alternatives to learn the R1-4, why did the author choose Cayley SGD method (Li et al., 2020).
>
> **A4:** We chose the Cayley SGD method for its efficiency in maintaining the orthonormality of the rotation matrix during optimization. Unlike naive methods (e.g., SGD) that fail to preserve orthonormality, i.e., orthonormality is a hard requirement otherwise the matrix is not a rotation matrix after optimization, and other costly orthonormal optimization techniques (e.g. [3] [4]] which require an expensive matrix inverse), Cayley SGD can efficiently optimize the orthonormal matrix.
> [3]  A riemannian conjugate gradient method for optimization on the stiefel manifold.
> [4]  A framework of constraint preserving update schemes for optimization on stiefel manifold
>
> **W5:** Why table 6 shows results with SpinQuant_hard even for W4A8?
>
> **A5:** We included the results for SpinQuant_hard in Table 6 to offer a comprehensive overview of the latency overhead associated with Hadamard rotation. This allows for a more complete understanding of the speed performance implications across different configurations.
>
>
> **W6:** What speedup benefit we may have with SpinQuant_hard and SpinQuant_easy with W4A4?
>
>
> **A6:** Thank you for the question. Below are the results from the W4A4 speed testing using the CUDA core implementation [5]
>
> | Layer Size | Batch Size | FP16 | W-int4-A-int4 | W-int4-A-int4 + Hadamard |
> | - | - | - | - | - |
> | 8192x8292 | 32 |  115.462ms | 31.693ms | 33.780ms |
>
> The W-int4-A-int4 configuration without Hadamard rotation achieves a 3.64x speed increase. With Hadamard rotation, the speed increase is 3.41x, indicating an approximate 6.6% overhead introduced by the rotation.
> This 6.6% overhead is consistent with the 8% latency overhead reported in the paper.
> It is important to note that these results are based on CUDA core implementations. Utilizing Tensor cores with operation fusion can yield greater speed improvements. However, this approach necessitates a more sophisticated design and implementation process, which typically involves longer design cycles. Given the promising accuracy results demonstrated by SpinQuant, we anticipate growing interest and investment in advanced W4A4 Tensor core kernels with Hadamard operations. This development is expected to further enhance W4A4 SpinQuant speed in the future. The primary aim of presenting the latency results in the SpinQuant paper is to demonstrate that the online Hadamard rotation incurs an acceptable level of overhead.
> [5] QuaRot: Outlier-Free 4-Bit Inference in Rotated LLMs
>
> **W7:** The contribution is primarily centered around rotation to get rid of activation outliers and smoothen out while for the quantization part the authors heavily rely on GPTQ (or RTN). The baseline GPTQ may often be considered a calibration heavy that requires Hessian information. I have concerns on the calibration overhead of the method as well as the novelty.
>
> **A7:** SpinQuant is fully compatible with both GPTQ and naive RTN. To isolate the contributions of GPTQ and rotation to the overall performance, we present results for SpinQuant combined with RTN in the W4A4KV16 quantization scenario.
>
>  | | LLaMA-3 8B |  | LLaMA-2 7B |  | LLaMA-2 13B |  | LLaMA-2 70B |  |
>  | - | - | - | - | - | - | - | - | - |
>  | Method | Zero-shot | Wiki | Zero-shot | Wiki | Zero-shot | Wiki | Zero-shot | Wiki |
>  | RTN | 43.9 | 2e2 | 35.6 | 2e3 | 35.3 | 7e3 | 35.1 | 2e5 |
>  | SmoothQuant | 40.3 | 9e2 | 41.8 | 3e2 | 44.9 | 34.5 | 57.7 | 57.1 |
>  | LLM-QAT | 44.9 | 42.9 | 47.8 | 12.9 | 34.3 | 4e3 | -- | -- |
>  | GPTQ | 40.6 | 2e2 | 36.8 | 9e3 | 35.2 | 5e3 | 35.5 | 2e6 |
>  | SpinQuant-hard RTN | 64.6 | 7.7 | 61.8 | 6.1 | 65.8 | 5.4 | 71.1 | 3.9 |
>  | SpinQuant-hard GPTQ | **65.8** | **7.1** | **64.1** | **5.9** | **67.2** | **5.2** | **71.0** | **3.8** |
>
> Our analysis indicates that the **primary accuracy gains are attributed to the incorporation of learned rotations**, which enhance accuracy by 6.5 ~ 20.9 percentage points over previous methods (including GPTQ). The subsequent integration of GPTQ further boosts performance by up to 2.3 percentage points. We added these results and discussions to the revision.
>
> Regarding calibration overhead, SpinQuant is on par with GPTQ, as previously noted in response to W3. Conceptually, the distinction is substantial. Note that most existing post-training quantization methods rely on calibration, such as AWQ, SmoothQuant, and GPTQ. While GPTQ employs the calibration set to compute the Hessian matrix and iteratively adjust for quantized weight errors, SpinQuant leverages the calibration set to optimize rotations which effectively mitigate outliers and quantization errors.

---

> ### Author Response · Authors · 2024-11-19
>
> **Questions:**
>
>
> **Q1:** It would be interesting to see the performance of such dynamic quantization (SpinQuant) with static quantization alternative [1]. How does SpinQuant compare w.r.t these alternative methods? [1][2]
> [1] PREFIXQUANT: STATIC QUANTIZATION BEATS DYNAMIC THROUGH PREFIXED OUTLIERS IN LLMS, arxiv.org/abs/2410.05265
> [2] TESSERAQ: ULTRA LOW-BIT LLM POST-TRAINING QUANTIZATION WITH BLOCK RECONSTRUCTION, arxiv.org/abs/2410.19103
>
> **A1:** In essence, spinquant would work just as well for static-quantization, as it would also benefit from reduced outliers. It would of course be valuable to compare the performance of SpinQuant with more recent studies [1][2]. However, both papers [1] and [2] were published on arXiv after the ICLR submission deadline. As a result, it was not feasible to anticipate these developments and include a comparison with these works in our submission.
>
> **Q2:** Is the R3, R4 needed in case someone uses block reconstruction of [2]?
>
> **A2:** R3 and R4 can be utilized with block reconstruction when targeting low-bit activation quantization. Since block reconstruction only differs from pre-layer reconstruction in terms of the loss used for reconstruction, it does not address outliers and can benefit from R3 and R4
>
> **Q3:** Why the choice of ``optimizing these rotation matrices on Stiefel manifold", is not fully clear.
>
> **A3:**  The Stiefel manifold, denoted as $ V_k(\mathbb{R}^n) $, is defined as the set of all orthonormal $k$-frames in $\mathbb{R}^n$. This implies that the Stiefel manifold $V_k(\mathbb{R}^n)$ includes all orthonormal matrices of size $ n \times k $. Rotation optimization should be conducted on the Stiefel manifold because the rotation matrix must meet the requirements of orthonormality. We will add these definitions and make it more clear in the paper.

---

> ### Author Response · Authors · 2024-11-25
>
> Dear Reviewer 6v9t,
>
> For your queries, we carefully prepared the answers. We look forward to you considering raising your score if our response effectively addressed your question.
>
> Thank you!
> Authors

---

> ### Comment · Reviewer_6v9t · 2024-11-26
> **Response to authors' rebuttal**
>
> Thanks to the authors for providing a detailed response that addresses most of my doubts and/or concerns. In summary, I am convinced on the utility of the hadamard rotation of both weight and activation that spinquant brought in. My only concern remains on the contribution, as the idea of rotation to mitigate the outlier issue has been explored earlier (I respect the authors on being extremely fair on the literature survey). Though making it efficient to merge and apply at network level reduces the rotation inefficiency, but again applying that for activation may induce additional inefficiency. Also, learning the rotation for activation, may ask to learn these things associated to different downstream tasks. For weight it might be one time. Did I miss something here? So I am still at the fences, I have increased my scores accordingly.

---

> > ### Author Response · Authors · 2024-11-27
> >
> > Thank you for your response and the follow-up questions!
> >
> > Regarding your first question about “additional inefficiencies from applying rotations to activations”, we would like to clarify that this is not the case with our approach. When we rotate the weight matrices, the output activations are inherently rotated as well. This means we do not introduce additional rotation matrices specifically for the activations. This is one of the key advantages of our algorithm and a significant distinction from the QuIP\# method. For instance, when applying the $R_1$ rotation, as illustrated in Figure 1, it is integrated into the embedding weight matrix ($W_{emb}' = W_{emb} @ R_1$). Consequently, the output activations from the embedding layer are also rotated ($X_{emb-out}'$  $= X_{emb-in} @ W_{emb}'$ $= X_{emb-in} @ W_{emb} @ R_1 $ $= X_{emb-out} @ R_1$). In this way, by applying $R_1$, the fully connected layers, including q-proj/k-proj/v-proj in attention and up-proj/gate-proj in FFN receive rotated activations, which become easy to quantize. In the meantime, as shown in Figure 1 (b) the reverse rotations $R_1^{-1}$ are absorbed in the corresponding weight matrices. Thus, no additional rotation matrices are needed for activation rotation in this case; it naturally results from the weight rotation. In contrast, QuIP\# uses $U^T (quantized(UWV^T)(V x))$ for every weight matrix, introducing hundreds of online rotations. We hope this explanation clarifies the process. We will also add these detailed explanations in our paper.
> >
> > As for your second question about learning rotations for activations across different downstream tasks, we do not optimize the rotation for each specific task. Instead, we use one calibration set, such as the Wiki dataset, across various downstream tasks. While we acknowledge that activations are influenced by input data, we have observed consistent patterns in activation distribution. This observation aligns with findings from SmoothQuant, which relies on such patterns to identify outliers using a calibration set and apply scaling to weights, thereby generalizing to other tasks. Empirically, we have found that our learned rotation, calibrated using the Wiki dataset, transfers effectively to new tasks, such as MMLU, TLDR9+. In our response to review U4Hi, we also demonstrated that SpinQuant significantly improves 5-shot accuracy on the MMLU benchmark and 1-shot ROUGE scores on the TLDR9 summarization benchmark.
> >
> > We hope these explanations address your inquiries. If you have any more questions, please let us know. We're always happy to answer them!

---

> > ### Public Comment · ~Tang_Bin1 · 2024-11-28
> > **Same Concern about the Contribution**
> >
> > Thank the reviewer for the insightful feedback and for highlighting this point. I appreciate the detailed discussion from both the reviewer and the authors.
> >
> > Upon further reflection, I noticed that DuQuant: Distributing Outliers via Dual Transformation Makes Stronger Quantized LLMs also leverages the concept of rotation to address the outlier issue. While the authors have conducted a commendable and thorough literature survey, I share the reviewer’s concern regarding the contribution of rotation to mitigate the outlier issue. I feel it might be worth discussing this aspect in the next vision.
> >
> > Once again, thank the reviewer for bringing this to light, and I look forward to seeing how the authors address this in their response or revision.

---

> ### Public Comment · ~Bohan_Zhuang1 · 2024-11-30
> **Clarification from public**
>
> I noticed that DuQuant was released on ArXiv after the release of SpinQuant.

---

> ### Author Response · Authors · 2024-12-02
> **Replying the public comment**
>
> Thanks for mentioning a relevant work. DuQuant uses permutation and rotation, while SpinQuant is the first paper to learn the rotation for better quantization accuracy. This contribution is acknowledged by Reviewers 6v9t, Jf43, U4Hi, tcZR, brBn.
>
> Besides, SpinQuant exhibits significant accuracy boost compared to DuQuant when quantizing a hard-to-quantize LLaMA-3 8B model to W4A4:
>
>  |  | PIQA(↑) | ARC-e(↑) | ARC-c(↑) | BoolQ(↑) | HellaS.(↑) | WinoG.(↑) | Avg.(↑) | Wiki2 (↓) |
>  |-|-|-|-|-|-|-|-|-|
>  | Full-Precision | 80.9 | 77.8 | 53.4 | 81.3 | 79.2 | 72.8 | 74.2 | 6.1 |
>  | DuQuant | 76.2 | 70.4 | 43.7 | 74.3 | 73.9 | 67.8 | 67.7 | 8.1 |
>  | SpinQuant | **77.5** | **75.0** | **50.9** | **78.9** | **75.9** | **68.5** | **71.1** | **7.1** |
>
> These accuracy improvements are attributed to learning the rotations, which is proposed by SpinQuant and not utilized by DuQuant. This further underscores SpinQuant's superiority and effectiveness.

---

> ### Author Response · Authors · 2024-12-03
>
> Dear Reviewer 6v9t,
>
> We hope our response has resolved your questions. If you have any further questions, we are here to answer them! Thanks again for the time and effort you have dedicated to reviewing our paper. If you think our paper has merit and our response addresses your concerns, we look forward to your positive feedback!
>
> Thanks,
> Authors

---

### Author Response · Authors · 2024-11-19

We sincerely thank the reviewers for their insightful feedback.

We are encouraged to see that all the reviewers acknowledged the paper's contribution in introducing an innovative approach through learned rotation to mitigate outliers in LLMs [Reviewers **6v9t, Jf43, U4Hi, brBn**], conducting thorough analysis and extensive experiments [Reviewers **6v9t, Jf43, tcZR**]; and demonstrating significant performance improvements over previous methods by a wide margin [Reviewers **6v9t, Jf43**].

We greatly appreciate the reviewers' efforts in evaluating our work and the constructive suggestions are invaluable for further improving the paper, for example, adding additional ablation studies on the calibration data [Reviewer **6v9t**], extending testing to more model architectures [Reviewer **U4Hi**], and providing further details on optimization time [Reviewer **brBn**]. These recommendations have been integrated into the revised manuscript. Below, we provide detailed point-by-point responses to each of the reviewers' comments.

---

### Meta-Review · Area_Chair_NaB8 · 2024-12-19

**Metareview:**

The paper introduces SpinQuant, a new method to learn rotation matrices for the purpose of improving quantized network accuracy by reducing errors from outliers in weight and activation distributions. Reviewers agree that the paper is well structured, clearly written, and that it presents a valuable contribution. The empirical performance of the method was well received. The raised weaknesses of the paper were its somewhat incremental novelty, given prior work with rotation ideas, the desire for more detailed corroboration in terms of more extensive benchmarks and e.g. compute time, as well as more theoretical analysis. The authors have added several clarifications, extensions and revisions to address a good amount of open concerns. The reviewers who engaged in discussion have acknowledged that these have improved the paper and have raised their recommendations to accept. In regards to the concern on incremental novelty, the AC  agrees with some reviewers that a thorough embedding in the literature should not be to the detriment of a paper. Overall, the AC believes that the advantages of the paper outweigh the remaining need for improvements, especially because the presentation of the paper is good. The AC thus recommends to accept the paper.

**Additional Comments On Reviewer Discussion:**

The authors have provided extensive replies and revisions to the paper, addressing several of the concerns raised by the reviewers throughout the discussion period. Most reviewers have acknowledged these responses and despite having some reservations left, e.g. whether the novelty is significant enough, agree that they improved the paper. Reviewer Jf43 has unfortunately not further engaged with the paper, but has originally voted to accept this paper. The AC believes this reviewer’s comments have also in parts been addressed. Reviewer 6v9t has raised their score from an initial score of 3 to a score of 5. Following the discussion it is unclear to the AC what remaining concerns the reviewer has that have in turn lead to a borderline reject score. Given that the other reviewers agree that the paper passes the threshold for acceptance, the AC is inclined to agree with them and votes to accept the paper.

---

### Decision · Program_Chairs · 2025-01-22

Accept (Poster)